# DRIFT-Net: A Spectral–Coupled Neural Operator for PDEs Learning

**Jiayi Li**
University of New South Wales
`jiayi.li18@student.unsw.edu.au`

**Flora D. Salim**[*]
University of New South Wales
`flora.salim@unsw.edu.au`

## Abstract

Learning PDE dynamics with neural solvers can significantly improve wall-clock efficiency and accuracy compared with classical numerical solvers. In recent years, foundation models for PDEs have largely adopted multi-scale windowed self-attention, with the scOT backbone in POSEIDON serving as a representative example. However, because of their locality, truly globally consistent spectral coupling can only be propagated gradually through deep stacking and window shifting. This locality can weaken globally consistent spectral coupling, which has been associated with increased error accumulation and drift during closed-loop rollouts. To address this, we propose **DRIFT-Net**. It employs a dual-branch design comprising a spectral branch and an image branch. The spectral branch is responsible for capturing global, large-scale low-frequency information, whereas the image branch focuses on local details and nonstationary structures. Specifically, we first perform controlled, lightweight mixing within the low-frequency range. Then we fuse the spectral and image paths at each layer via bandwise weighting, which avoids the width inflation and training instability caused by naive concatenation. The fused result is transformed back into the spatial domain and added to the image branch, thereby preserving both global structure and high-frequency details across scales. Compared with strong attention-based baselines, DRIFT-Net achieves lower error and higher throughput with fewer parameters under identical training settings and budget. On Navier–Stokes benchmarks, the relative $L_1$ error is reduced by 7%–54%, the parameter count decreases by about 15%, and the throughput remains higher than scOT. Ablation studies and theoretical analyses further demonstrate the stability and effectiveness of this design. The code is available at `https://github.com/cruiseresearchgroup/DRIFT-Net`.

## 1 Introduction

Partial differential equations (PDEs) underpin science and engineering. Repeated high-accuracy simulations remain costly at scale (Trefethen, 2000; Benner et al., 2015). Neural operators address this challenge by learning mappings directly between function spaces. This enables fast inference across resolutions and inputs and supports cross-mesh generalization (Kovachki et al., 2023). Representative models include the Fourier Neural Operator (FNO) (Li et al., 2021) and DeepONet (Lu et al., 2021). Building on these advances, PDE foundation models adopt multi-scale windowed self-attention. POSEIDON with its scOT backbone is a representative example.

Nevertheless, windowed self-attention is local. Global dependencies emerge only gradually with depth and shifted windows. This weakens *globally consistent spectral coupling* and can induce error accumulation and drift during closed-loop autoregressive rollouts (Lippe et al., 2023). In practice, naive cross-scale or cross-branch concatenation inflates channel width and destabilizes training. Purely spectral operators are global but often overemphasize low-frequency structure and underfit nonstationary local details.

**Our approach.** We introduce DRIFT-Net, a dual-branch neural operator. The spectral branch performs controlled low-frequency global mixing. The image branch handles local interactions. The

---

[*]Corresponding author.

two branches are fused bandwise through a non-expansive mechanism. This avoids width inflation and preserves high-frequency detail. Implementation details, including the low-frequency mixing and bandwise fusion, are presented in §4.

**Contributions.**

- **Modular operator unit.** DRIFT-NET provides a dual-branch unit with controlled low-frequency mixing and bandwise non-expansive fusion. It enhances global coupling, local-detail fidelity, and training stability. The unit can be swapped in for windowed self-attention blocks in multi-scale operator backbones.

- **Performance and efficiency.** Under matched training schedules and hardware, DRIFT-NET reduces final-time relative $L_1$ error by 7% to 54% on Navier–Stokes benchmarks. It uses about 15% fewer parameters and achieves higher throughput than scOT. See §5.

- **Mechanism and reusability.** Ablations and spectral analyses show how non-expansive fusion and controlled low-frequency mixing support stable training and improved generalization. The design is reusable and modular for stronger PDE foundation models.

## 2 RELATED WORK

**Neural operators.** Neural operators learn mappings between function spaces and approximate solution operators of PDEs (Li et al., 2021; Kovachki et al., 2023). A recent survey categorizes existing architectures into three main types: deep operator networks (DeepONets), integral-kernel neural operators, and transformer-based neural operators (Liu et al., 2025). DeepONet adopts a branch–trunk factorization and comes with operator-approximation guarantees (Lu et al., 2021). Integral-kernel neural operators model nonlocal interactions through parameterized kernels. FNO implements the kernel operator in the Fourier domain (Li et al., 2021), and variants such as Geo-FNO and U-FNO extend this idea to irregular geometries and multiphase flow (Li et al., 2023; Wen et al., 2022); CNO and neural operators with localized integral or differential kernels further emphasize multi-scale structure and local operators (Raonić et al., 2023; Liu-Schiaffini et al., 2024). Transformer-based neural operators treat grid points as tokens and use self-attention to approximate data-dependent kernels, and are widely used in multi-task and pretraining settings (McCabe et al., 2023; Hao et al., 2024; Yang et al., 2023; Rahman et al., 2024).

**PDE foundation models and multi-scale attention.** A recent line of work builds PDE foundation models on top of transformer-based neural-operator backbones. Examples include MPP (a spatio-temporal transformer surrogate trained autoregressively on multiphysics datasets), the operator transformer DPOT with Fourier attention, and attention-based neural operators such as ICON and CoDA-NO; all of them perform large-scale pretraining on multi-dataset or multiphysics corpora and transfer to downstream PDE tasks (McCabe et al., 2023; Hao et al., 2024; Yang et al., 2023; Rahman et al., 2024). Within multi-scale attention frameworks, POSEIDON employs a multi-scale windowed attention operator transformer (scOT) with a U-Net hierarchy and time-conditioned normalization, and shows strong cross-task generalization on Navier–Stokes benchmarks (Herde et al., 2024; Liu et al., 2021; 2022). Windowed self-attention computes attention efficiently in local windows, but global dependencies are established only gradually through depth and window shifts, which can weaken global spectral coupling and contribute to rollout drift at long horizons (Lippe et al., 2023).

**Positioning.** We propose DRIFT-Net as a spectral–spatial coupled integral-kernel neural-operator backbone, which in the above taxonomy falls into the class of integral-kernel neural operators. Architecturally, DRIFT-Net is composed of stacked DRIFT blocks: each block augments the multi-scale attention backbone with an explicit spectral path for controlled low-frequency mixing and a non-expansive bandwise fusion mechanism that integrates the spectral and image branches without increasing the feature width. Methodological details and quantitative comparisons are given in Sections 4 and 5.

## 3 PROBLEM SETUP

**Problem formulation.** We consider a generic time-dependent partial differential equation on a spatial domain $D \subset \mathbb{R}^d$ and time horizon $T > 0$:

$$
\begin{aligned}
\partial_t u(x,t) + \mathcal{L}\big(u, \nabla_x u, \nabla_x^2 u, \dots\big) &= 0, \qquad \forall\, x \in D \subset \mathbb{R}^d,\ t \in (0, T), \\
\mathcal{B}(u) &= 0, \qquad \forall\, (x,t) \in \partial D \times (0, T), \\
u(x, 0) &= a(x), \quad \forall\, x \in D.
\end{aligned}
\tag{3.1}
$$

Here $L$ and $B$ denote the differential and boundary operators, and $a(x)$ is the initial datum. Time-independent problems are also covered by this formulation. If a steady state exists, the long-time limit $(t \to \infty)$ yields the steady PDE

$$
L\big(u(x), \nabla_x u(x), \nabla_x^2 u(x), \dots\big) = 0, \qquad B(u) = 0, \qquad x \in D,
\tag{3.2}
$$

which is the time-independent counterpart of equation 3.1.

**Solution operator.** Let $X$ denote the state space, for example a suitable function space on $D$. The solution can be described by a map $S : [0, T] \times X \to X$ such that, for any $t \in [0, T]$ and $a \in X$, $u(t) = S(t, a)$. Equivalently, for each fixed $t$ we define the flow map $S_t : X \to X$ with $S_t(a) = S(t, a)$.

**Underlying operator learning task.** Given a distribution $\mu$ over initial conditions $a \in X$, the goal is to learn an approximate solution operator $S^\star$ that closely reproduces the true operator $S$. For any $a \sim \mu$, the learned operator should generate the trajectory $\{S^\star(t, a)\}_{t \in [0,T]}$ that approximates $\{S(t, a)\}_{t \in [0,T]}$ for all $t$. The model is expected to produce the time evolution from $a$, given boundary conditions, without relying on intermediate information, analogous to a classical solver.

**Learning objective.** On a discrete grid of size $H \times W$ with $C$ components, let $u_t \in \mathbb{R}^{C \times H \times W}$ denote the state at discrete time $t$. We learn a one-step operator $F_\theta : u_t \mapsto u_{t+1}$ with teacher forcing and a relative $L_p$ objective with $p \in \{1, 2\}$. At test time, $F_\theta$ is composed autoregressively to produce a full trajectory $\{\hat{u}_t\}_{t=1}^T$. Details of the training schedule, data splits, rollout horizon, and evaluation metrics are specified in Sec. 5.

## 4 METHOD

Existing neural operators suffer from weak global coupling and the loss of high-frequency detail in long-horizon prediction (Lippe et al., 2023). We propose DRIFT-NET, a U-Net–style encoder–decoder (Ronneberger et al., 2015) with two parallel branches: a frequency path for cross-scale global interaction and an image path for local, nonstationary structures (Wen et al., 2022). At each scale, we fuse the branches by inverse-transforming the frequency output to the spatial domain and adding it to the image output. The model hinges on three mechanisms: (1) controlled low-frequency mixing to strengthen long-range dependencies without disturbing high-frequency modes; (2) bandwise fusion with radial gating for smooth cross-band transitions (Rahaman et al., 2019; Xu et al., 2020); and (3) a frequency-weighted loss to counter spectral bias. We first outline the architecture and then detail each component.

### 4.1 ARCHITECTURE OVERVIEW

DRIFT-NET follows a hierarchical encoder–decoder and augments each scale with an explicit spectral path (Fig. 1). In the *image branch*, ConvNeXt-style blocks extract local and nonstationary structures; down/up-sampling is realized by patch merging/expansion to form a U-shaped hierarchy. In parallel, the *spectral branch* converts features at the same scale to the Fourier domain via rFFT2, applies a controlled transformation only on low-frequency modes (Sec. 4.2), and fuses them back by a smooth bandwise mechanism (Sec. 4.3). After iFFT2, the spectral output is *added* to the image branch feature at that scale.

Two design choices are key. First, the spectral path provides *immediate* global receptive fields at every resolution, so globally consistent coupling does not rely on deep stacking or large kernels. Second, the additive cross-branch fusion is *non-expansive* in width (no concatenation), which avoids

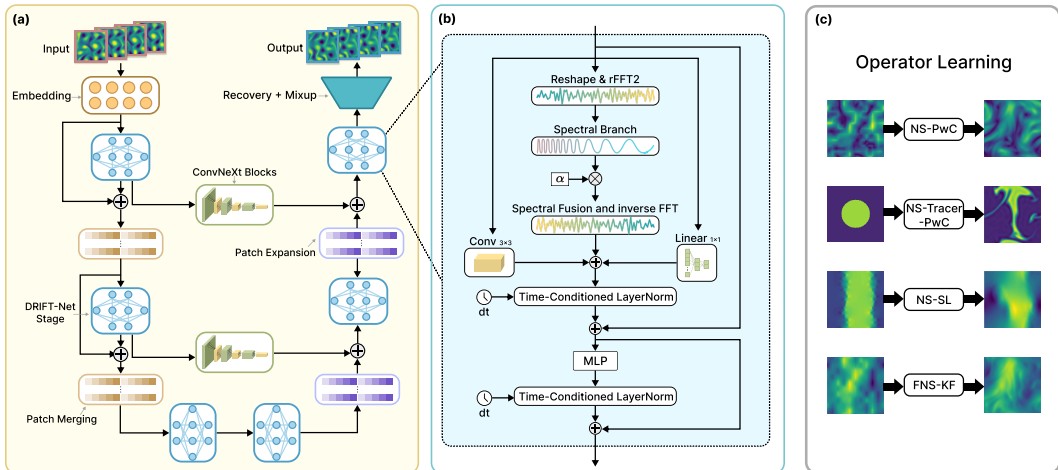

Figure 1: **Overall architecture of DRIFT-NET.** (a) U-Net style encoder–decoder with ConvNeXt blocks and patch merging/expansion. (b) Spectral branch (rFFT2, spectral fusion with radial gating) fused additively with image branch via inverse FFT. (c) Example operator-learning tasks: NS-PwC, NS-Tracer-PwC, NS-SL, FNS-KF.

channel inflation and empirically stabilizes optimization. Across scales, the network thus propagates coarse global structure while preserving high-frequency details refined by local convolutions. This decomposition mirrors the physical intuition that low-$k$ modes control large-scale dynamics, whereas high-$k$ modes encode fine structures, discontinuities, and small eddies. For the architecture pseudocode content, please refer to Appendix E.

## 4.2 CONTROLLED LOW-FREQUENCY MIXING

In complex physical scenarios, achieving global-range feature interaction is crucial for capturing long-range correlated dynamic behavior. However, indiscriminately applying global convolutions or frequency-domain operations across the entire frequency spectrum may overly amplify high-frequency noise and disrupt local details, leading to model instability. Therefore, the frequency-domain branch of DRIFT-NET performs controlled global mixing operations only on low-frequency modes, focusing global coupling on large-scale structures. Low-frequency components represent the field's overall shape and long-term evolution trends and play a decisive role in the global dynamics. In contrast, high-frequency components carry fine-grained local structures. By introducing global mixing only in the low-frequency band, the model can effectively propagate large-scale information at each layer to directly couple distant spatial locations, while maximally preserving high-frequency details and avoiding unnecessary interference with small-scale structures.

**Fourier-domain decomposition and motivation.** At the feature level, low spatial frequencies correspond to smooth, domain-wide basis functions whose coefficients modulate global structures; high frequencies capture localized variations. Modifying only a bounded set of low-$k$ coefficients yields a global yet parsimonious interaction pattern that respects the separation of scales. This aligns with classical spectral methods and with the observation that neural networks tend to fit low-frequency components first (spectral bias) (Rahaman et al., 2019; Xu et al., 2020). Our design *uses* this bias constructively by letting the network explicitly *learn* how to mix low-$k$ channels, while leaving high-$k$ content intact.

**Block computation.** In each DRIFT-NET block, we apply a 2D real FFT (rFFT2) to the input feature $X_{\text{in}}$ to obtain its frequency-domain representation $\hat{X}(k)$; by Hermitian symmetry of real-valued inputs, only half of the spectrum needs to be represented. We then use a *learnable rectangular* low-frequency mask $M_{\text{low}}(k_x, k_y)$ to split the spectrum into a low-frequency part $\hat{X}_{\text{low}}$ and a high-frequency residual $\hat{X}_{\text{high}} = \hat{X} - \hat{X}_{\text{low}}$. Within the low-frequency range $k \in M_{\text{low}}$, we introduce a learnable channel-wise complex linear transformation $W$ (acting per frequency, without cross-

frequency coupling) to mix the corresponding spectral coefficients (Rao et al., 2021):

$$\hat{V}(k) = W\,\hat{X}(k)\,, \quad k \in M_{\text{low}}\,,$$
$$\hat{V}(k) = \hat{X}(k)\,, \quad k \notin M_{\text{low}}\,. \tag{4.1}$$

For later use we denote $\hat{V}_{\text{low}}(k) = \mathbb{1}_{k \in M_{\text{low}}}\hat{V}(k)$ and keep $\hat{X}_{\text{high}}(k) = \hat{X}(k) - \mathbb{1}_{k \in M_{\text{low}}}\hat{X}(k)$. In practice, $W$ is unconstrained (no explicit spectral-norm clipping), which keeps the transform expressive. We use rFFT2/iFFT2 and the inverse transform yields a real-valued field by construction (Cooley & Tukey, 1965; Frigo & Johnson, 2005). This design focuses global mixing on large scales while leaving high-frequency content intact. For implementation details, refer to Appendix C.

## 4.3 Bandwise Fusion with Radial Gating

After the above low-frequency mixing, we need to fuse the modified low-frequency information from the frequency branch with the complementary content to produce the final spectrum for the inverse transform back to the spatial domain. If one simply performs a hard band replacement or a direct addition, discontinuities may occur at the frequency band boundaries or amplitude overshoot may be introduced in certain bands. Therefore, we design a smooth and stable frequency-band fusion mechanism that uses a radial gating coefficient varying with frequency to weight and combine the two frequency-domain signals.

Specifically, we assign each frequency $k$ in the spectrum a weight $\alpha(k) \in [0,1]$ to balance the contributions of the low-frequency mapped component and the high-frequency residual. We compute $\alpha(k)$ as a function of the frequency magnitude (radial frequency) using lightweight per-band processing and expand it to per-frequency weights so that in the low-frequency region $\alpha(k) \approx 1$ (favoring the mixed global component), whereas in the high-frequency region $\alpha(k) \approx 0$ (preserving fine local details). The fusion is then computed as

$$\hat{Y}(k) = \alpha(k)\,\hat{V}_{\text{low}}(k) + \big(1 - \alpha(k)\big)\,\hat{X}_{\text{high}}(k)\,, \tag{4.2}$$

and since $\alpha(k) \in [0,1]$, by convexity we have the pointwise magnitude bound

$$\big|\hat{Y}(k)\big| \leq \max\Big\{\,\big|\hat{V}_{\text{low}}(k)\big|,\ \big|\hat{X}_{\text{high}}(k)\big|\,\Big\}\,. \tag{4.3}$$

This bandwise "clamping" effect avoids introducing energy larger than either source at any frequency and empirically stabilizes training. Finally, we apply iFFT2 to obtain the spectral-path output in the spatial domain and add it to the image branch at the same scale.

**Why radial and why additive.** Choosing $\alpha(k)$ as a radial function enforces isotropy in fusion and prevents directional artifacts around the mask boundary. The convex combination above yields a non-expansive operator in the frequency domain (no overshoot), which directly translates to fewer ringing artifacts after iFFT2. In the spatial domain, we merge the spectral and image signals by *addition* rather than concatenation. This preserves feature dimensionality and makes the fusion behave as a residual correction that injects globally coupled information while letting the image branch keep full control over high-frequency refinement. See Appendix C for details.

## 4.4 Frequency-Weighted Loss

Despite the above architectural improvements in global coupling and local detail fidelity, the model training stage still needs to address the issue of spectral bias. Conventional losses tend to be dominated by low-frequency errors during optimization, causing the model to preferentially fit large-scale structures while converging slowly on smaller-amplitude yet physically important high-frequency details. As a result, fine structures in the predicted field may be underfitted and gradually become blurred over time.

To mitigate spectral bias in a simple and stable manner, we add a *frequency-weighted* auxiliary term in the Fourier domain. Let $E = u_\theta - u$ be the prediction error and $\hat{E}$ its rFFT2 under the same normalization. We reweight the error spectrum by a radial weight $w(r) \propto r^\alpha$ (with $r$ the normalized frequency magnitude) and minimize

$$L = L_{\text{base}} + \lambda\,\mathbb{E}\big[\,w(r)\,\big|\hat{E}(k)\big|^2\,\big]\,, \tag{4.4}$$

where $L_{\text{base}}$ is the standard $L_p$ loss and $(\lambda, \alpha)$ are scalars. This radial weighting increases sensitivity to high-$|k|$ components and complements the bandwise fusion used in the frequency path, thereby improving the fidelity of multi-scale structures in long-horizon predictions (Fuoli et al., 2021; Jiang et al., 2021). From a functional-analytic perspective, this amounts to emphasizing higher-order (roughness-related) components of the error—akin to moving from a pure $L_p$ metric toward a Sobolev-like metric—so that small-scale discrepancies are not overshadowed by low-frequency energy during optimization. Practically, tuning $\lambda$ controls the balance between coarse and fine accuracy, while $\alpha$ modulates how aggressively high frequencies are emphasized; we find moderate values suffice to counter underfitting of fine structures without degrading large-scale fidelity. See Appendix B for details.

## 5 EXPERIMENTS

**Protocol and metrics.** We follow the POSEIDON evaluation protocol (Herde et al., 2024). All models, including DRIFT-NET and all baselines, use the same train/validation/test splits and pre-processing, and are trained under a shared data budget and training protocol: the same number of supervised trajectories per task, identical epoch budgets, and a common optimizer and learning-rate schedule inherited from the POSEIDON scOT configuration (see Appendix F.4). On a discrete grid with $C$ channels and $H \times W$ spatial points, we train a one-step operator $F_\theta : u_t \mapsto u_{t+1}$ with single-step teacher forcing. At test time, we apply $F_\theta$ autoregressively in a closed loop to reach the common target time $T^*$. We report the test-set mean relative $L^1$ error at the final time:

$$e_{\text{rel-L1}} = \frac{\|\hat{u}_{T^*} - u_{T^*}\|_1}{\|u_{T^*}\|_1}, \qquad \|v\|_1 := \frac{1}{CHW} \sum_{c=1}^{C} \sum_{i=1}^{H} \sum_{j=1}^{W} |v_{c,i,j}|.$$

For tasks with multiple quantities of interest (QoIs)—for example, NS-Tracer-PwC predicts $(u_x, u_y, c)$ while NS-PwC, NS-SL, and FNS-KF output only $(u_x, u_y)$—we compute the relative error for each QoI and then take an unweighted average:

$$e_{\text{task}} = \frac{1}{|Q|} \sum_{q \in Q} \frac{\|\hat{u}_{T^*}^{(q)} - u_{T^*}^{(q)}\|_1}{\|u_{T^*}^{(q)}\|_1}.$$

**Baselines.** We compare DRIFT-NET with scOT (Herde et al., 2024), FNO (Li et al., 2021), U-NO (Rahman et al., 2022), a U-Net–ConvNeXt baseline, and a Refiner U-Net baseline adapted from PDE-Refiner (Lippe et al., 2023); see Appendix for detailed configurations.

**Benchmark tasks.** We evaluate DRIFT-NET on four canonical unsteady Navier–Stokes benchmarks from the POSEIDON suite (Herde et al., 2024). NS-SL (shear layer) tests vortex roll-up and mixing from a perturbed interface. NS-PwC (piecewise-constant vorticity) stresses the advection of sharp discontinuities. NS-Tracer-PwC extends NS-PwC by introducing a passive scalar $c$, which requires accurate coupling between the velocity and tracer fields. FNS-KF (forced Kolmogorov flow) sustains two-dimensional turbulence via steady forcing, challenging long-horizon stability and multi-scale fidelity.

To cover different types of PDEs, we additionally include three tasks: Poisson–Gauss describes a stationary Poisson equation with Gaussian forcing and assesses performance on elliptic problems. The Allen-Cahn equation (ACE) models interface evolution and phase separation, representing a prototypical reaction-diffusion process. Wave-Gauss simulates two-dimensional wave propagation in a Gaussian medium and tests the modeling of hyperbolic wave phenomena.

In addition to these four benchmarks, we also consider two ApeBench-generated forced Kolmogorov variants constructed using a high-accuracy pseudo-spectral solver (Koehler et al., 2024). These two variants share the same PDE setup and preprocessing but differ in physical parameters: one is more turbulent, whereas the other is comparatively smoother. To emphasize long-horizon robustness, we evaluate on both variants with a closed-loop rollout length of $T = 100$ and, for these two datasets, compare DRIFT-NET against scOT under identical settings.

**Main results.** Table 1 reports the final-time relative $L^1$ errors across all seven PDE benchmarks under the matched training protocol described above. On the four POSEIDON Navier–Stokes tasks, DRIFT-NET consistently attains the lowest error among all compared architectures, reducing the error of scOT by between $7\%$ and $54\%$ while using fewer parameters. Among the baseline models, scOT is typically the strongest, whereas FNO underperforms on most unsteady flow tasks.

On all three additional PDE benchmarks (Poisson–Gauss, Allen–Cahn, and Wave–Gauss), DRIFT-NET achieves the best test error, with modest gains on Poisson–Gauss and more pronounced improvements on Allen–Cahn and Wave–Gauss. These results indicate that replacing windowed-attention blocks with DRIFT blocks improves accuracy not only on Navier–Stokes benchmarks but also on elliptic, parabolic, and hyperbolic PDEs.

Table 1: Final-time relative $L^1$ error (lower is better). Closed-loop rollouts to the common target time $T^*$ on all benchmarks. Values are *test-set means*. **Best in bold**.

| Task | scOT | FNO | U-NO | U-Net–ConvNeXt | Refiner U-Net | DRIFT-NET |
|---|---|---|---|---|---|---|
| NS-SL | 3.96 | 3.69 | 3.57 | 3.89 | 3.48 | **3.40** |
| NS-PwC | 2.35 | 4.57 | 1.62 | 2.22 | 1.95 | **1.09** |
| NS-Tracer-PwC | 5.18 | 9.46 | 10.15 | 9.63 | 4.59 | **4.19** |
| FNS-KF | 4.65 | 4.43 | 11.70 | 5.94 | 4.38 | **4.32** |
| Poisson–Gauss | 0.87 | 1.29 | 1.76 | 1.92 | – | **0.86** |
| Allen–Cahn | 1.10 | 1.53 | 1.27 | 1.61 | 1.11 | **1.03** |
| Wave–Gauss | 12.34 | 16.97 | 13.94 | 16.02 | 13.33 | **12.15** |

**Long-horizon behavior.** Although final-time metrics are informative, a full closed-loop trajectory over a long horizon provides additional insight into cumulative drift. We therefore analyse two *ApeBench*-generated forced Kolmogorov variants constructed with a high-accuracy pseudo-spectral solver (Koehler et al., 2024). Both datasets share the same PDE setup but differ in physical parameters, with one being more turbulent and the other comparatively smoother. For both variants we run closed-loop rollouts up to $T=100$ and compare DRIFT-NET against scOT under identical settings.

Table 2 summarizes the final-time relative $L^1$ error at $T=100$. On the turbulent variant, DRIFT-NET attains a lower final error than scOT (110.87 vs. 114.14). On the smoother variant, the improvement is more pronounced (57.17 vs. 62.99), indicating increased robustness across different flow regimes.

Table 2: Forced Kolmogorov flows from ApeBench. Final-time relative $L^1$ error (lower is better) at $T=100$. Values are *test-set means*. **Best in bold**.

| Task | scOT | DRIFT-NET |
|---|---|---|
| KF-Long (turbulent, ApeBench, $T=100$) | 114.14 | **110.87** |
| KF-Long (smoother, ApeBench, $T=100$) | 62.99 | **57.17** |

Beyond the final-time value, we also consider the mean relative $L^1$ error across the rollout and a linear growth slope obtained by a least-squares fit of error versus time. On the turbulent variant, DRIFT-NET attains a mean error of 69.89 compared with 74.76 for scOT, and an error-growth slope of 1.154 compared with 1.185 for scOT. On the smoother variant, the mean error is 31.07 compared with 35.41 for scOT, and the slope is 0.581 compared with 0.641 for scOT. Figure 2 shows the corresponding error–time curves, which remain consistently lower for DRIFT-NET with slightly flatter tails at large $t$.

Finally, to better visualise how errors accumulate, we plot in Figure 3 the instantaneous error growth on the smoother Kolmogorov variant, defined as the difference between consecutive relative $L^1$ values. DRIFT-NET exhibits smaller instantaneous growth than scOT for most of the rollout, especially in the early and mid-range time steps, and only converges to similar values near the end of the horizon. This pattern is consistent with the design goal of DRIFT blocks, namely strengthening global low-frequency coupling while maintaining high-frequency fidelity and avoiding rapid error amplification.

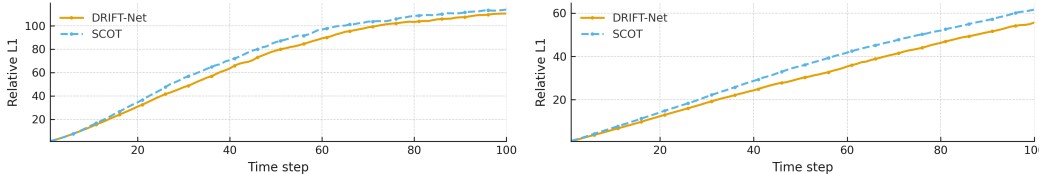

Figure 2: **Error vs. time on ApeBench-based long-horizon Kolmogorov datasets** ($T$=100). Left: turbulent; Right: smoother. Solid line: DRIFT-NET; dashed line: scOT.

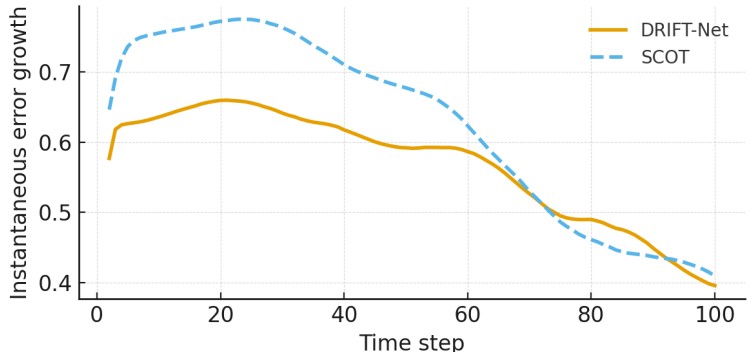

Figure 3: **Instantaneous error growth on the smoother ApeBench Kolmogorov dataset.** We plot the per-step change in relative $L^1$ error over a rollout of length $T$=100. Solid line: DRIFT-NET; dashed line: scOT.

**Efficiency comparison.** We compare the computational cost of scOT and DRIFT-NET on FNS-KF (Table 3). Under identical hardware and inference settings, the $1.0\times$ DRIFT-NET uses 17M vs. 20M parameters for scOT, reduces peak training memory from 17.25 GB to 10.87 GB, shortens training time per epoch from 17.30 to 14.91 minutes, and increases inference throughput from 118 to 158 steps/s. At $0.5\times$ width, both models become cheaper, and DRIFT-NET still has lower memory and higher throughput (224 vs. 175 steps/s). Overall, on FNS-KF, DRIFT-NET is both more accurate and more efficient than scOT.

Table 3: Computational cost on FNS-KF. Parameter count, peak training memory, training time per epoch, and inference throughput (higher is better) for scOT and DRIFT-NET at two width settings. Measurements are taken under identical inference and training settings.

| Model | Params (M) | Peak train mem (GB) | Train time / epoch (min) | Throughput (steps/s) |
|---|---|---|---|---|
| scOT | 20 | 17.25 | 17.30 | 118 |
| scOT-0.5× | 10 | 8.58 | 15.81 | 175 |
| DRIFT-NET | 17 | 10.87 | 14.91 | **158** |
| DRIFT-NET-0.5× | 9 | 6.82 | 13.96 | **224** |

**Ablation studies.** We assess the contribution of each major component of DRIFT-NET: Low-Frequency Mixing (LFM), Radial Gating (RG), Frequency-Weighted Loss (FWL), and the two operators in the image branch, namely the $3\times3$ depth-wise convolution and the $1\times1$ pointwise linear layer. We train ablated variants on NS-PwC and NS-Tracer-PwC, holding all other hyperparameters fixed and using the same training budget and evaluation protocol as the full model. Removing LFM or RG noticeably worsens final-time accuracy; removing FWL also degrades performance, although to a smaller degree. Dropping the $3\times3$ depth-wise convolution or the $1\times1$ pointwise linear layer further increases the error, showing that the ConvNeXt-style image branch is also important (Table 4). Concretely, relative to the full model, removing LFM increases the final error by 0.56 and 2.13 on NS-PwC and NS-Tracer-PwC, respectively; removing RG increases it by 0.61 and 2.36; removing

FWL yields smaller increases of 0.27 and 1.17; removing the depth-wise convolution adds 0.02 and 0.17; and removing the pointwise linear layer adds 0.22 and 0.02.

Table 4: Ablation of DRIFT-NET components. Numbers are final-time relative $L^1$ errors (lower is better) on NS-PwC and NS-Tracer-PwC under the same training and evaluation protocol. **Best in bold**.

| Model variant | NS-PwC | NS-Tracer-PwC |
|---|---|---|
| Full DRIFT-NET (ours) | **1.09** | **4.19** |
| w/o Low-Frequency Mixing (LFM) | 1.65 | 6.32 |
| w/o Radial Gating (RG) | 1.70 | 6.55 |
| w/o Frequency-Weighted Loss (FWL) | 1.36 | 5.36 |
| w/o 3×3 depth-wise conv | 1.11 | 4.36 |
| w/o 1×1 pointwise linear | 1.31 | 4.21 |

**Spectral analysis for ablations.** To characterize scale-dependent effects, we analyze the evolution of bandwise errors over time (normalized RMSE per frequency band). Figure 4 compares four model variants (the full DRIFT-NET, no FWL, no RG, and no LFM), each showing error-versus-time curves for multiple wavenumber bands. The full model most effectively suppresses error growth in the mid- and high-frequency bands (those with $k \geq 16$). In contrast, removing RG leads to the earliest and most pronounced increase in the highest-frequency band. Removing FWL yields a marked late-stage increase in high-frequency error. Removing LFM increases both mid- and high-frequency errors, consistent with weakened low-frequency global coupling.

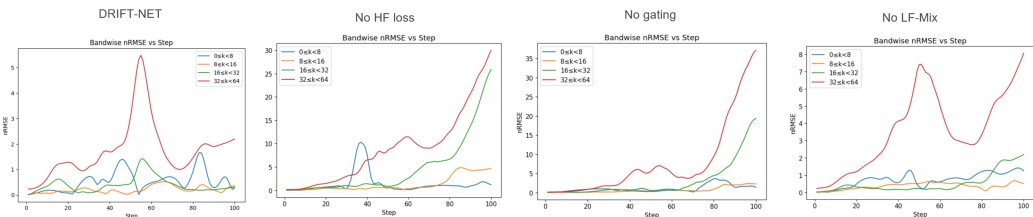

Figure 4: **Bandwise nRMSE vs. step for ablations.** Left to right: full DRIFT-NET, no HF loss, no gating, no LF-Mix.

## 6 LIMITATIONS AND FUTURE WORK

DRIFT-Net has limitations. It uses a mask to select low frequencies. The initial band is hand-tuned. The mask learns weights later, but the cutoff can be task specific. The model applies FFTs in many blocks. This adds memory traffic and latency on very large grids. Our tests target two-dimensional flow. Three-dimensional flow and coupled multi-physics may bring new training issues and higher cost. Future work will learn spectral partitions end to end. We will study 3D and multi-physics PDEs. We will pair the model with adaptive resolution and mesh refinement. We will also test irregular domains and complex boundary conditions.

## 7 CONCLUSION

We introduced DRIFT-NET, a spectral–spatial coupled integral-kernel neural-operator backbone. Each DRIFT block combines a spectral branch with controlled low-frequency mixing and a ConvNeXt-style image branch, and fuses them via bandwise radial gating in a non-expansive way. A frequency-weighted loss further mitigates spectral bias. On four POSEIDON Navier–Stokes benchmarks, DRIFT-NET reduces final-time relative $L^1$ error by $7\%$–$54\%$ compared to scOT, while using about $15\%$ fewer parameters and achieving higher inference throughput. Additional results on Poisson–Gauss, Allen–Cahn, and Wave–Gauss show that these gains extend beyond Navier–Stokes

to elliptic, parabolic, and hyperbolic PDEs. Because DRIFT blocks are modular and architecture-agnostic, they can replace windowed attention blocks in existing multi-scale operator backbones and may be integrated into future PDE foundation models.

## ACKNOWLEDGMENTS

This project is supported by the ARC Centre of Excellence for Automated Decision-Making and Society (CE200100005).

This research includes computations using the computational cluster Katana supported by Research Technology Services at UNSW Sydney.

We also thank Hira Saleem for helpful discussions, feedback, and support during this project; her contributions are gratefully acknowledged.

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

APPENDIX

# A  THEORETICAL PROOFS AND BOUNDS

## A.1  SETUP AND NOTATION

Let $\Omega = \mathbb{T}^2$ be a periodic domain. Denote by $\mathcal{F}$ the 2-D rFFT with forward normalization (`norm=forward`). For a real-valued tensor $u \in \mathbb{R}^{H \times W \times C}$, write $\hat{u} = \mathcal{F}(u)$. Two learnable Nyquist-scaled cut-offs $\kappa_x, \kappa_y \in (0, 0.5]$ define a low-frequency rectangle

$$\hat{u}_{\text{low}} = \hat{u} \cdot \mathbf{1}_{\{\, |k_x| < \kappa_x,\ |k_y| < \kappa_y \,\}}, \qquad \hat{u}_{\text{high}} = \hat{u} - \hat{u}_{\text{low}},$$

with area $\sigma_{\text{LF}} = \kappa_x \kappa_y \leq 0.25$. The spectral half-plane is partitioned into concentric radial bands $\{B_j\}_{j=0}^{J-1}$ for gating.

## A.2  PER-BLOCK LIPSCHITZ UPPER BOUND

**Core DRIFT operator.**  Inside the low rectangle a shared complex matrix $W \in \mathbb{C}^{C \times C}$ mixes channels:

$$\hat{u}_{\text{low-mix}}(k) = W \, \hat{u}_{\text{low}}(k), \quad k \in M_{\text{low}}.$$

Let the three parallel branches of a DRIFT layer be: (i) low-band mixer $S$, (ii) depth-wise $3 \times 3$ convolution $C$, (iii) $1 \times 1$ linear map $L$. With forward-normalized FFT, Parseval on the low rectangle yields a fixed scaling absorbed into operator constants.

**Lemma A.1 (Core DRIFT operator bound).**  For the residual-free core $R_{\text{DRIFT}} = S + C + L$,

$$\|R_{\text{DRIFT}}\|_{\text{Lip}} \leq \sqrt{\sigma_{\text{LF}}} \, \rho_W + K_{\text{conv}} + K_{\text{lin}},$$

where $\rho_W = \|W\|_2$, and $K_{\text{conv}}, K_{\text{lin}}$ are operator norms of the depth-wise conv and the $1 \times 1$ linear, respectively. If the block is wrapped by an outer residual $I + R_{\text{DRIFT}}$, then $\|I + R_{\text{DRIFT}}\|_{\text{Lip}} \leq 1 + \|R_{\text{DRIFT}}\|_{\text{Lip}}$.

**Swin-style reference bound.**  A Swin-style block with identity shortcut decomposes as $x \mapsto x + A_{\text{attn}}(x) + A_{\text{mlp}}(x)$, hence

$$\|F_{\text{SWIN}}\|_{\text{Lip}} \leq 1 + \|A_{\text{attn}}\|_2 + \|A_{\text{mlp}}\|_2.$$

## A.3  RELATIVE NETWORK-LEVEL BOUND

**Proposition A.2 (Tighter cumulative bound).**  Suppose both DRIFT and Swin-style blocks use the same outer residual $I + \cdot$. If for every depth

$$\sqrt{\sigma_{\text{LF}}} \, \rho_W + K_{\text{conv}} + K_{\text{lin}} < \|A_{\text{attn}}\|_2 + \|A_{\text{mlp}}\|_2,$$

then for any number of layers $L$,

$$\prod_{\ell=1}^{L} \|F_{\text{DRIFT}}^{(\ell)}\|_{\text{Lip}} < \prod_{\ell=1}^{L} \|F_{\text{SWIN}}^{(\ell)}\|_{\text{Lip}}.$$

Consequently, DRIFT-Net admits a strictly smaller worst-case gain than an equally deep Swin-style stack without requiring either network to be contractive ($< 1$).

**Discrete Grönwall implication.**  Let $e_{m+1} \leq \bar{K}_\theta \, e_m + \eta_m$ with one-step defect $\eta_m$. Replacing $\bar{K}_\theta^{\text{SWIN}}$ by $\bar{K}_\theta^{\text{DRIFT}} < \bar{K}_\theta^{\text{SWIN}}$ flattens the geometric factor in the discrete Grönwall inequality:

$$e_m \leq \bar{K}_\theta^{\,m} e_0 + \frac{1 - \bar{K}_\theta^{\,m}}{1 - \bar{K}_\theta} \, \bar{\eta}, \qquad \bar{\eta} = \max_{i < m} \eta_i.$$

## A.4  RADIAL-BAND GATING IS ENERGY NON-EXPANSIVE

**Lemma A.3 (Pointwise amplitude bound).**  For any fixed $\alpha \in [0, 1]$ and Fourier location $k$,

$$\hat{v}(k) = \alpha \, \hat{u}_{\text{low-mix}}(k) + (1 - \alpha) \, \hat{u}_{\text{high}}(k) \Rightarrow |\hat{v}(k)| \leq \max\{\, |\hat{u}_{\text{low-mix}}(k)|,\ |\hat{u}_{\text{high}}(k)| \,\}.$$

*Proof.* Convexity: $|\alpha a + (1 - \alpha) b| \leq \alpha |a| + (1 - \alpha)|b| \leq \max\{|a|, |b|\}$. $\qquad \square$

**Remark (input-dependent gates).** When $\alpha$ depends on the input via a band-statistics MLP, the amplitude bound still holds pointwise; the Lipschitz constant additionally includes a term from $\partial\alpha/\partial u$. In practice, one may detach gradients through $\alpha$ in the stability analysis or constrain the gate MLP by spectral normalization.

### A.5 OPERATOR-NORM CONTROLS FOR CONV/LINEAR

**Depth-wise $3 \times 3$ conv.** On a periodic grid, the depth-wise conv is block-circulant and diagonalized by the DFT. Hence $\|T_k\|_2 = \max_{h,w} |\widehat{k}_{h,w}|$. Ensuring $\sum_{i,j} |k_{ij}| \leq 1$ or projecting to $\max_{h,w} |\widehat{k}_{h,w}| \leq 1$ yields $K_{\text{conv}} \leq 1$.

$1 \times 1$ **linear.** If $W \in \mathbb{R}^{C \times C}$ is orthogonally initialized and projected each step to the spectral ball of radius $c_L \leq 1$, then $\|W\|_2 \leq c_L$, giving $K_{\text{lin}} \leq c_L$.

## B SOBOLEV-WEIGHTED ONE-STEP DEFECT BOUND

**Theorem B.1 (Sobolev-closed defect).** Fix $s > 0$, $\lambda_{\text{hf}} > 0$ and define $L_{\text{Sob}}(u, \hat{u}) = \lambda_{\text{hf}}\|\Lambda^s(u - \hat{u})\|_2^2$ with $\Lambda^s = (I - \Delta)^{s/2}$. If $\mathbb{E}\, L_{\text{Sob}} \leq \varepsilon$, then the expected one-step defect $\eta_m = \|\, \hat{u}_{m+1} - \Phi_\theta(\hat{u}_m)\,\|_2$ satisfies

$$\mathbb{E}\,\eta_m \leq \lambda_{\text{hf}}^{\frac{1}{2+s}}\, \varepsilon^{\frac{1}{2+s}} =: \bar{\eta}.$$

*Proof sketch.* Parseval (with `norm=forward`) gives $L_{\text{Sob}} = \lambda_{\text{hf}} \sum_k (1 + \|k\|^2)^s |e_k|^2$, $e_k = u_k - \hat{u}_k$. Split the spectrum at radius $r$: $\eta_m^2 = \sum_{\|k\| \leq r} |e_k|^2 + \sum_{\|k\| > r} |e_k|^2$; the HF term $\leq (1 + r^2)^{-s}\lambda_{\text{hf}}^{-1}L_{\text{Sob}}$. Optimizing $r$ yields the stated bound; insert into the discrete Grönwall inequality.$\square$

## C SPECTRAL PIPELINE DETAILS

**Low/high split (implementation).** Given input size $(H, W)$, the rFFT has size $(H_{\text{fft}}, W_{\text{fft}}) = (H, W/2 + 1)$. Two learnable scalars $\kappa_x = \sigma(\theta_x)$, $\kappa_y = \sigma(\theta_y)$ define

$$M_{\text{low}} = \{\, |k_x| < \kappa_x H_{\text{fft}},\ |k_y| < \kappa_y W_{\text{fft}} \,\}, \quad \hat{u}_{\text{low}} = \hat{u} \cdot \mathbf{1}_{M_{\text{low}}}, \quad \hat{u}_{\text{high}} = \hat{u} - \hat{u}_{\text{low}}.$$

**Radial gate (energy-fraction driven).** Let $r_{ij} = \sqrt{(i/(H_{\text{fft}} - 1))^2 + (j/(W_{\text{fft}} - 1))^2}$ and bands $B_j = \{(i,j) : \lfloor Jr_{ij} \rfloor = j\}$, $j = 0, \ldots, J - 1$. Define a high-frequency energy fraction

$$E_{\text{HF}}(k) = \frac{|\hat{u}_{\text{high}}(k)|^2}{|\hat{u}_{\text{low}}(k)|^2 + |\hat{u}_{\text{high}}(k)|^2 + \varepsilon}.$$

Average within each band to obtain $\bar{f}_{n,c}(j)$, pass through a two-layer MLP with sigmoid output, and broadcast back: $\alpha_{n,c}(k) = \sigma(\text{MLP}(\bar{f}_{n,c}(\lfloor Jr_{ij} \rfloor))) \in (0, 1]$.

**Spectral fusion and inference-only taper.** Blend as

$$\hat{v}(k) = \alpha(k)\, \hat{u}_{\text{low-mix}}(k) + \big(1 - \alpha(k)\big)\, \hat{u}_{\text{high}}(k), \qquad v = \mathcal{F}^{-1}(\hat{v}).$$

At evaluation time, a lightweight outer-band taper may be applied on the outermost ring: $\hat{v}(k) \leftarrow (1 - \beta\, \bar{\alpha}(k))\, \hat{v}(k)$ with $\beta \in (0, 1/2]$ and $\bar{\alpha}$ the channel-wise mean gate. Because $0 \leq \alpha \leq 1$ and the taper factor $\leq 1$, every Fourier mode is non-expansive.

## D COMPLEXITY AND THROUGHPUT PROTOCOL

**Asymptotic costs per DRIFT block.** rFFT/iFFT pair: $\mathcal{O}(HW \log(HW))$; band statistics & broadcast: $\mathcal{O}(HW)$; low-band mixing: $\mathcal{O}(|M_{\text{low}}| C^2)$ with $|M_{\text{low}}| \ll HW$. Thus DRIFT-Net matches the asymptotic complexity of window attention while eliminating dense projections in attention blocks.

# E  PSEUDOCODE

---

**Algorithm 1** DRIFT-NET one-step forward and training loss (code-aligned)

---

**Require:** current field $u_t \in \mathbb{R}^{C \times H \times W}$; #scales $L$; learnable LF mask params $(k_x, k_y)$; complex mixers $\{W_\ell\}_{\ell=1}^{L}$; radial band gates $\{\text{BandGate}_\ell\}_{\ell=1}^{L}$; base loss $L_{\text{base}}$; spectral weights $w(r)$; coefficient $\lambda$

1:

**Ensure:** predicted next field $\hat{u}_{t+1}$; loss $L$ (if training)

 2:  $x \leftarrow \text{Embed}(u_t)$
 3:  **for** $\ell = 1$ to $L$ **do**
 4:      $\hat{X} \leftarrow \text{rFFT2}(x)$
 5:      $(H_{\text{fft}}, W_{\text{fft}}) \leftarrow \text{shape}(\hat{X})$;  $k_x \leftarrow \lfloor \sigma(\theta_x)\, H_{\text{fft}} \rfloor$, $k_y \leftarrow \lfloor \sigma(\theta_y)\, W_{\text{fft}} \rfloor$
 6:      $\hat{X}_{\text{low}} \leftarrow \hat{X} \odot \mathbf{1}_{[:k_x, :k_y]}$;  $\hat{X}_{\text{high}} \leftarrow \hat{X} - \hat{X}_{\text{low}}$
 7:      $\hat{V}_{\text{low}}(k) \leftarrow W_\ell\, \hat{X}(k)$ for $k \in [:k_x, :k_y]$;  $\hat{V}_{\text{low}}(k) \leftarrow 0$ otherwise
 8:      $\text{feat} \leftarrow \big||\hat{X}_{\text{high}}| - |\hat{X}_{\text{low}}|\big|$
 9:      $\alpha(k) \leftarrow \text{BandGate}_\ell\big(\text{Pool}_r(\text{feat}), \|k\|\big)$
10:      $\hat{Y}(k) \leftarrow \alpha(k)\, \hat{V}_{\text{low}}(k) + \big(1 - \alpha(k)\big)\, \hat{X}_{\text{high}}(k)$
11:      $y_{\text{spec}} \leftarrow \text{iFFT2}(\hat{Y})$
12:      $y_{\text{local}} \leftarrow \text{DWConv}_\ell(x) + \text{PointwiseLinear}_\ell(x)$
13:      $z \leftarrow \text{Norm}_\ell\big(y_{\text{spec}} + y_{\text{local}}, \text{time}\big)$
14:      $x \leftarrow x + z$
15:      **if** $\ell < L$ **then**
16:          $x \leftarrow \text{Downsample}(x)$
17:  **for** $\ell = L$ down to 1 **do**
18:      **if** $\ell < L$ **then**
19:          $x \leftarrow \text{Upsample}(x)$
20:      $x \leftarrow \text{ConvNeXtBlock}_\ell(x)$
21:  $\hat{u}_{t+1} \leftarrow \text{Recover}(x)$
22:  **if** training **then**
23:      $E \leftarrow \hat{u}_{t+1} - u_{t+1}$;  $\hat{E} \leftarrow \text{rFFT2}(E)$
24:      $L_{\text{base}} \leftarrow \|E\|_p$                                             $\triangleright\, p \in \{1, 2\}$
25:      $L_{\text{freq}} \leftarrow \lambda \cdot \mathbb{E}_k\big[w(\|k\|)\, |\hat{E}(k)|^2\big]$
26:      $L \leftarrow L_{\text{base}} + L_{\text{freq}}$
27:  **return** $\hat{u}_{t+1}$ (and $L$ if training)

---

# F  EXPERIMENTAL SETUP AND HYPERPARAMETERS

## F.1  OVERALL PROTOCOL

We follow the POSEIDON downstream evaluation protocol (Herde et al., 2024) and use the public Poseidon datasets: NS-SL, NS-PwC, NS-Tracer-PwC, FNS-KF, Allen–Cahn (ACE), Wave–Gauss, and Poisson–Gauss. All fields are provided on a fixed $128 \times 128$ Cartesian grid; we train and evaluate directly on this grid without any additional spatial interpolation or resampling.

For all time-dependent tasks (NS-SL, NS-PwC, NS-Tracer-PwC, FNS-KF, ACE, Wave–Gauss), each trajectory consists of a sequence of states $\{u_t\}_{t=0}^{N_t}$ sampled at a fixed time step. We train a one-step operator

$$F_\theta : u_t \mapsto u_{t+1},$$

acting on grids in $\mathbb{R}^{C \times 128 \times 128}$, where $C$ is the number of physical channels (e.g., $C=2$ for velocity, $C=3$ for velocity plus tracer). Training uses single-step teacher forcing: we sample a time index $t$ from each trajectory, feed $u_t$ into the model, and minimize the one-step prediction error to $u_{t+1}$. In Wave–Gauss the spatially varying wave speed is provided as an additional static input channel.

Poisson–Gauss is time-independent; there we directly learn the mapping from source field $f(x, y)$ to steady-state solution $u(x, y)$ on the same grid.

At test time we use the learned one-step map in closed loop. Given an initial state $u_0$ we generate an autoregressive rollout $\{\hat{u}_t\}_{t=1}^{T^*}$ by iteratively applying $F_\theta$ and compare the final prediction $\hat{u}_{T^*}$ with the ground truth $u_{T^*}$ at the standard target snapshot $T^*$ used in POSEIDON for each task. For Poisson–Gauss we evaluate directly on the steady-state solution.

Unless otherwise stated, we report the mean test relative $L_1$ error at the evaluation time: for a single-quantity task,

$$\text{rel-}L_1 = \frac{\|\hat{u}_{T^*} - u_{T^*}\|_1}{\|u_{T^*}\|_1}, \quad \|v\|_1 = \tfrac{1}{CHW} \sum_{c,i,j} |v_{cij}|,$$

averaged over all test trajectories. For tasks with multiple quantities of interest (e.g., NS-Tracer-PwC), we compute the relative error per quantity and take an unweighted average. All models (DRIFT-Net and baselines) use the same train/validation/test splits and the same number of training trajectories as in the Poseidon releases.

### F.2 DOWNSTREAM DATASETS AND TASKS

We evaluate DRIFT-NET and all baselines on seven downstream tasks from the Poseidon collection and on two additional long-horizon Kolmogorov-flow benchmarks generated with APEBench. All datasets use a fixed $128 \times 128$ Cartesian grid.

**NS-SL.** NS-SL is a two-dimensional incompressible Navier–Stokes benchmark with double shear-layer initial conditions. The Poseidon NetCDF file provides a single variable `velocity` with shape $40000 \times 21 \times 2 \times 128 \times 128$ (trajectories $\times$ time steps $\times$ channels $\times H \times W$), where the two channels are horizontal and vertical velocity. Trajectories are simulated on the unit square up to $T = 1$ with 21 uniformly spaced snapshots; the official split is $39640/120/240$ trajectories for train/validation/test.

**NS-PwC and NS-Tracer-PwC.** NS-PwC and NS-Tracer-PwC share the same Poseidon dataset. The NetCDF file contains a `velocity` variable of shape $20000 \times 21 \times 3 \times 128 \times 128$. The three channels correspond to horizontal velocity, vertical velocity, and a passive tracer convected by the flow. We treat the first two channels as the NS-PwC task (predicting only the velocity field) and all three channels as NS-Tracer-PwC (velocity plus tracer). All trajectories are simulated up to $T = 1$ with 21 snapshots, and we use the standard $19640/120/240$ train/validation/test split.

**FNS-KF.** FNS-KF is a forced incompressible Navier–Stokes benchmark with piecewise constant vorticity initial conditions and steady Kolmogorov forcing. The dataset provides a single variable `solution` with shape $20000 \times 21 \times 2 \times 128 \times 128$ for the two velocity components. Simulations run on the unit square up to $T = 1$ with 21 snapshots and use a time- and sample-independent forcing field $f(x, y) = 0.1 \sin(2\pi(x + y))$. We adopt the official $19640/120/240$ split.

**ACE.** ACE contains trajectories of the Allen–Cahn reaction–diffusion equation. The NetCDF file has a single variable `solution` of shape $15000 \times 20 \times 128 \times 128$, representing a scalar concentration field. The equation is solved on the unit square up to $T = 2 \times 10^{-4}$ with 20 uniformly spaced snapshots. We use the default $14700/60/240$ trajectories for training, validation, and testing.

**Wave–Gauss.** Wave–Gauss is a second-order wave equation with spatially varying wave speed. The dataset provides two variables: `solution` of shape $10512 \times 15 \times 128 \times 128$ for the scalar wave field, and `c` of shape $10512 \times 128 \times 128$ for the static wave-speed field. We treat `c` as an additional static input channel that is concatenated to the dynamic field at every time step. Trajectories are simulated on the unit square up to $T = 1$ with 15 snapshots, and the official split is $10212/60/240$ trajectories.

**Poisson–Gauss.** Poisson–Gauss is a time-independent benchmark based on the Poisson equation with Gaussian source terms. The NetCDF file contains two variables of shape $20000 \times 128 \times 128$: `source`, the right-hand side $f(x, y)$, and `solution`, the corresponding steady-state solution

$u(x, y)$. We learn the operator mapping $f \mapsto u$ directly on the $128 \times 128$ grid, using the default split of $19640/120/240$ samples.

**ApeBench Kolmogorov flows.** For long-horizon evaluation we generate two Kolmogorov-flow datasets with APEBench. Both are based on the periodic 2D vorticity formulation of the incompressible Navier–Stokes equations with drag and single-mode Kolmogorov forcing,

$$\omega_t + \mathbf{u} \cdot \nabla \omega = \nu \Delta \omega - \gamma \omega + f(x, y), \qquad \mathbf{u} = (\partial_y \psi, -\partial_x \psi), \ \Delta \psi = \omega,$$

discretized with a pseudo-spectral solver on a $128 \times 128$ grid. We consider two parameter sets:

- **KF-Long (turbulent).** Corresponds to the NS-2D-Forced `BASE` configuration with $\nu = 10^{-2}$, $\gamma = -0.1$, forcing mode $k_{\text{force}} = 4$, and forcing scale $1.0$.
- **KF-Long (smoother).** Corresponds to the NS-2D-Forced `VARB` configuration with the same forcing mode but higher viscosity and drag: $\nu = 1.5 \times 10^{-2}$, $\gamma = -0.15$, $k_{\text{force}} = 4$, and forcing scale $1.0$.

For each configuration we procedurally generate $4,000$ trajectories with the APEBench code and split them into $3,700$ training, $200$ validation, and $100$ test trajectories. Unless otherwise stated, long-horizon results on these benchmarks are reported for closed-loop rollouts of length $T = 100$ time steps.

### F.3 MODEL CONFIGURATIONS AND PARAMETER COUNTS

All models are instantiated in a "base" configuration on $128 \times 128$ grids with comparable capacity. DRIFT-NET has about 17M parameters, while all baselines are scaled to about 20M parameters.

**DRIFT-NET.** DRIFT-NET is a four-level U-Net encoder–decoder with feature widths $(d_0, d_1, d_2, d_3) = (d, 2d, 4d, 4d)$ and two DRIFT blocks per resolution. Each block contains an image branch (ConvNeXt-style block with depthwise $3 \times 3$ convolution, pointwise $1 \times 1$ convolution, MLP, LayerNorm, and GELU) and a spectral branch that applies a 2-D FFT, restricts mixing to a learnable low-frequency mask, and maps back with an inverse FFT. Radial-band gating partitions Fourier modes into $J$ radial bands and uses a small MLP per band to blend the image and spectral features. We choose $d$ such that the total parameter count is $\approx 17M$ on $128 \times 128$; the "$0.5 \times$" variant used in the FNS-KF ablation halves all channel widths.

**scOT.** The scOT baseline follows the official POSEIDON implementation (code: `https://github.com/camlab-ethz/poseidon`): a multiscale Swin-style operator transformer with ConvNeXt bridges and time-conditioned layer norms (Herde et al., 2024). We use the *Poseidon-T* (scOT-T) configuration for all from-scratch runs, i.e., the tiny model used as the backbone of Poseidon, which has about 20M parameters. We keep the architectural hyperparameters (number of levels, Swin blocks per level, window size, patch size) identical to the released scOT-T config and only adapt the input/output channel dimensions to each task.

**FNO.** Our Fourier Neural Operator baseline follows Li et al. (Li et al., 2021) and uses the 2-D time-dependent FNO architecture from the official codebase (`https://github.com/li-Pingan/fourier-neural-operator`). The network consists of a lifting layer that maps the physical input channels to a width of $d_{\text{fno}} = 96$, followed by $L = 5$ Fourier layers with $m_x = m_y = 16$ retained complex Fourier modes per spatial dimension, and a projection layer back to the physical channels. We keep $L$ and $(m_x, m_y)$ fixed across all tasks and only tune $d_{\text{fno}}$ to control capacity; with $d_{\text{fno}} = 96$ the model has about 19M parameters on $128 \times 128$ grids. All other architectural choices (complex Fourier weights, FFT-based convolutions, and nonlinearity placement) follow the official implementation.

**U-NO.** The U-NO baseline uses the official U-shaped Neural Operator implementation (`https://github.com/ashiq24/UNO`) (Rahman et al., 2022). We employ a simplified 2-D UNO configuration with two resolution levels. At the finest resolution an operator block maps the physical channels to $\text{width}_{\text{uno}}$, we then downsample by a factor of two and apply an operator block with output width $2\,\text{width}_{\text{uno}}$, and finally upsample and fuse encoder and decoder features with an operator block

that maps $3\,\text{width}_{\text{uno}}$ back to $\text{width}_{\text{uno}}$ before a $1 \times 1$ convolution to the output channels. We set $\text{width}_{\text{uno}} = 88$ and use $m_x = m_y = 16$ Fourier modes in each operator block. This yields a total parameter count of about 20M on $128 \times 128$ grids.

**U-Net–ConvNeXt.** The U-Net–ConvNeXt baseline shares the same four-level encoder–decoder hierarchy as DRIFT-NET (down/up-sampling operators and skip connections), but contains only a spatial branch. We use the `UNet2d` backbone from PDEBench with ConvNeXt-style blocks (code: `https://github.com/pdebench/PDEBench`) and set the base width to `init_features` $= 50$, which determines the channel widths at each level via the standard U-Net doubling rule. With this setting the model has approximately 20M parameters on $128 \times 128$ grids. We do not include any spectral path or radial-band gating.

**Refiner U-Net.** The Refiner U-Net baseline is adapted from PDE-Refiner (`https://phlippe.github.io/PDERefiner/`) (Lippe et al., 2023). We use a modern four-level convolutional U-Net backbone with channel widths $(48, 96, 192, 384)$ and two residual $3 \times 3$ convolutional blocks per level, similar to the Unetmod backbone employed for Kolmogorov flow in PDE-Refiner. On top of this backbone we apply three refinement steps, each implemented as a lightweight residual update on the current prediction; all refinement steps share the same U-Net parameters. This configuration yields a total parameter count of approximately 20M on $128 \times 128$ grids.

## F.4 TRAINING AND OPTIMIZATION

All models are trained from scratch on each downstream task using the official Poseidon train/validation/test splits. For time-dependent datasets we train a one-step map $F_\theta : u_t \mapsto u_{t+1}$ with single-step teacher forcing and evaluate closed-loop rollouts as described in Appendix F.1. After each epoch we save a checkpoint and report test metrics for the checkpoint with the lowest validation relative $L_1$ error.

**Optimizer and schedule.** For all architectures (DRIFT-Net and all baselines) we use the same optimizer and learning-rate schedule, following the from-scratch scOT setup in POSEIDON (Herde et al., 2024; Li et al., 2021). Specifically, we use AdamW with an initial learning rate $3 \times 10^{-4}$, a cosine decay schedule with warm-up, and weight decay $10^{-6}$. These hyperparameters are fixed across all models and datasets and are not tuned per architecture; they were originally chosen for scOT rather than for DRIFT-NET.

**Epoch budgets and batch size.** On each task all models are trained for the same fixed number of epochs: 20 epochs on NS-SL, 100 epochs on Poisson–Gauss, and 40 epochs on all other Poseidon and ApeBench benchmarks. We do not use early stopping; the best-validation checkpoint is selected from these fixed budgets. Unless otherwise stated we use a mini-batch size of 40 trajectories for all models and all datasets, matching the batch size used for scOT and FNO in POSEIDON.

**Hardware and precision.** All experiments are run on a single NVIDIA H200 GPU in full single precision (FP32) without mixed-precision or low-precision training. Random seeds are fixed for each run so that our results are reproducible given the released code and configuration.

## G ADDITIONAL EXPERIMENTS

In this section we collect additional experiments that were used to address reviewer questions in the rebuttal but are not shown in the main text.

## G.1 WIDTH SCALING ON FNS-KF

To study the effect of model capacity, we train scOT and DRIFT-NET on FNS-KF with two width factors. Table 5 reports the parameter counts and test relative $L_1$ errors.

Reducing the width by a factor of two degrades performance for both models, but DRIFT-NET consistently achieves lower error than scOT at the same width (both at $0.5\times$ and $1.0\times$).

Table 5: Width scaling on FNS-KF.

| Model | Width factor | Params (M) | FNS-KF (rel-$L_1$) |
|---|---|---|---|
| scOT-0.5× | 0.5× | 10 | 5.98 |
| scOT-1.0× | 1.0× | 20 | 4.65 |
| DRIFT-NET-0.5× | 0.5× | 9 | 5.96 |
| DRIFT-NET-1.0× | 1.0× | 17 | 4.32 |

## G.2 SENSITIVITY TO LEARNING RATE AND SCHEDULE

We next study the sensitivity of scOT and DRIFT-NET to the base learning rate and the learning-rate schedule on FNS-KF. We train both models with AdamW for 40 epochs under a small grid of base learning rates $\{1 \times 10^{-4}, 3 \times 10^{-4}, 1 \times 10^{-3}\}$ and two schedules (cosine decay and step decay). Results are shown in Tables 6 and 7.

Table 6: FNS-KF: sensitivity to base learning rate under cosine schedule.

| Model | LR | Schedule | FNS-KF (rel-$L_1$) |
|---|---|---|---|
| scOT | $1 \times 10^{-4}$ | cosine | 4.78 |
| scOT | $3 \times 10^{-4}$ | cosine | 4.65 |
| scOT | $1 \times 10^{-3}$ | cosine | 4.75 |
| DRIFT-NET | $1 \times 10^{-4}$ | cosine | 4.45 |
| DRIFT-NET | $3 \times 10^{-4}$ | cosine | 4.32 |
| DRIFT-NET | $1 \times 10^{-3}$ | cosine | 4.42 |

Table 7: FNS-KF: sensitivity to base learning rate under step schedule.

| Model | LR | Schedule | FNS-KF (rel-$L_1$) |
|---|---|---|---|
| scOT | $1 \times 10^{-4}$ | step | 4.83 |
| scOT | $3 \times 10^{-4}$ | step | 4.70 |
| scOT | $1 \times 10^{-3}$ | step | 4.80 |
| DRIFT-NET | $1 \times 10^{-4}$ | step | 4.50 |
| DRIFT-NET | $3 \times 10^{-4}$ | step | 4.38 |
| DRIFT-NET | $1 \times 10^{-3}$ | step | 4.48 |

Across this grid, performance varies only moderately with the base learning rate and schedule, and DRIFT-NET consistently outperforms scOT in all tested configurations.

## G.3 SAMPLE EFFICIENCY ON FNS-KF

To characterise sample efficiency, we train DRIFT-NET on FNS-KF with different numbers of supervised training trajectories while keeping the model and optimisation settings fixed. Table 8 reports the test relative $L_1$ error as a function of the number of training trajectories.

As expected, test error decreases monotonically with the number of training examples, and DRIFT-NET remains stable as the data budget varies.

## G.4 CROSS-RESOLUTION BEHAVIOUR ON FNS-KF

To directly assess cross-resolution behaviour, we train DRIFT-NET on FNS-KF at $128 \times 128$ and evaluate the same model, without retraining, at $64 \times 64$, $96 \times 96$, and $128 \times 128$. Inputs and outputs are mapped between resolutions using standard interpolation operators. Table 9 shows the resulting test relative $L_1$ errors.

Table 8: FNS-KF: sample efficiency of DRIFT-NET.

| # training trajectories | DRIFT-NET (rel-$L_1$) |
|---|---|
| 2.5k | 6.25 |
| 5k | 5.48 |
| 10k | 4.96 |
| 20k | 4.32 |

Table 9: FNS-KF: cross-resolution evaluation of DRIFT-NET.

| Resolution | DRIFT-NET (rel-$L_1$) |
|---|---|
| $64 \times 64$ | 4.328 |
| $96 \times 96$ | 4.331 |
| $128 \times 128$ | 4.324 |

The errors are nearly identical across resolutions, indicating that in this experimental setup DRIFT-NET exhibits approximately mesh-agnostic behaviour.

### G.5 RESOLUTION SCALING WITH FFTS

Finally, we study how the computational cost of DRIFT-NET scales with spatial resolution on the ApeBench Kolmogorov-flow benchmark. We train DRIFT-NET on $128 \times 128$ and $256 \times 256$ grids with the same batch size and optimisation settings and measure the time per training step and peak GPU memory usage on a single H200. Table 10 summarises the results.

Table 10: Resolution scaling of DRIFT-NET on ApeBench Kolmogorov.

| Resolution | Time / step (ms) | Peak GPU memory (GB) |
|---|---|---|
| $128 \times 128$ | 114 | 11.7 |
| $256 \times 256$ | 387 | 23.8 |

Runtime and memory both increase smoothly with resolution and remain well within the capacity of a single H200 in the resolution regime considered in our experiments.

## H FORCED KOLMOGOROV FLOW (KF) VISUALIZATION

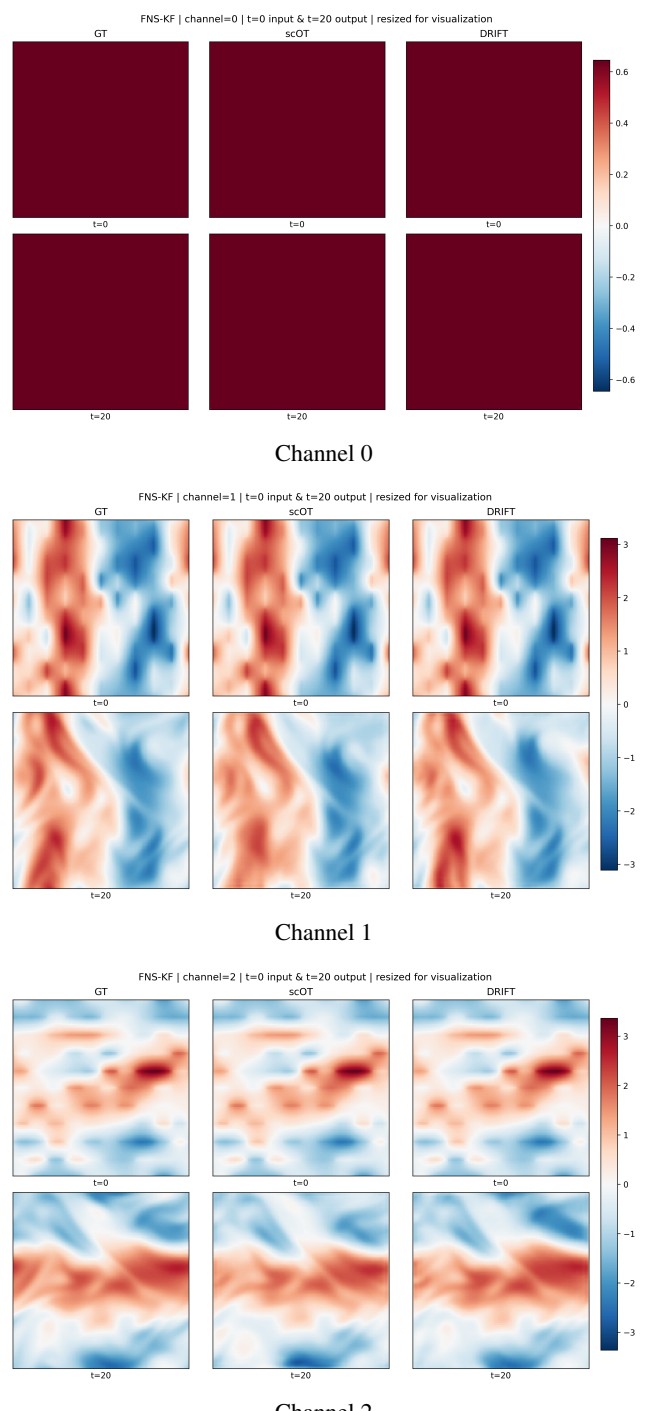

Figure 5: **FNS-KF qualitative visualization.** Each panel shows GT / scOT / DRIFT at $t=0$ (top) and $t=20$ (bottom) for a single channel. Example is illustrative (not a main result).

