# OpenReview forum: "DRIFT-Net: A Spectral-Coupled Neural Operator for PDEs Learning"
_ICLR.cc/2026/Conference — ICLR 2026 Poster_

### Official Review · Reviewer_RWgr · 2025-10-30

**Soundness:** 2
**Presentation:** 2
**Contribution:** 1
**Rating:** 2
**Confidence:** 4

**Summary:**

The paper introduces DRIFT-NET, a neural PDE solver aimed at solving the problem of error accumulation and "drift" observed in long-term, autoregressive predictions. The authors identify the cause of this drift in the use of local window attention in popular attention-based models, which only allow slow propagation of global information through the layers. To this end, DRIFT-NET proposes a frequency-weighted loss and a dual-branch architecture within a U-Net style backbone: an image branch using standard ConvNeXt blocks to capture local details and a spectral branch to capture global information. These branches are fused using learned low-frequency mixing and a band-wise radially-varying gate to combine low- and high-frequency components. DRIFT-NET is benchmarked against scOT and FNO on several 2D Navier-Stokes tasks, where it can improve performance with a lower parameter count and faster inference.

**Strengths:**

- The paper targets a well-known and critical failure mode of neural PDE solvers: long-horizon rollout drift.
- While the dual-branch approach with spectral mixing is not new, the *fusion mechanism* using band-wise radial gating seems to be an effective new component.
- The model demonstrates consistent performance improvement over the scOT and FNO baselines. In particular, it reduces accumulation errors in the rollouts with fewer parameters and a higher throughput. Moreover, an ablation study provides insights into the importance and effect of each of the three main components (LFM, RG, and FWL).

**Weaknesses:**

- While the paper claims to provide a modular and reusable design for PDE foundation models that “can be swapped in for windowed self-attention blocks in multi-scale operator backbones”, the effect of DRIFT-Net is only evaluated for a single backbone and only on variants of 2D Navier-Stokes equations. The spectral/image split is strongly motivated by fluid dynamics (large-scale flow vs. small-scale eddies), but may not be as optimal for other systems (e.g., hyperbolic equations, reaction-diffusion systems). Other PDE foundation models are not considered or mentioned, e.g. https://arxiv.org/pdf/2310.02994, https://arxiv.org/pdf/2403.03542, https://arxiv.org/abs/2304.07993, and https://arxiv.org/abs/2403.12553. To judge the benefit of the DRIFT-Net module, one needs to evaluate its effect on other types of PDEs, with other backbones, and also other dimensions (since, as mentioned in the limitations, the improvements in throughput might not hold anymore when using 3D FFTs). In the current form, there is also not enough evidence to claim that the drift during rollouts is *directly* caused by windowed self-attention.
- It would be important to compare the performance and throughput to adapted versions of scOT/FNO with a comparable parameter count. Moreover, it is important to note that the performance of (multi-scale) windowed self-attention heavily depends on the underlying kernel and hardware optimizations used for the attention module.
- Strictly speaking, the proposed architecture does not seem to be a neural operator (in the definition of https://arxiv.org/abs/2108.08481), since it is not discretization-invariant, i.e. it does not generalize to resolutions other than the training resolutions.
- DRIFT-Net is only compared to two baselines and no details are provided on the FNO baseline. While some variants of FNO are mentioned in the related works (e.g., U-NO and AFNO), they are not compared against and there exist further FNO versions that bear similarity with DRIFT-Net (e.g. https://arxiv.org/pdf/2109.03697 and https://arxiv.org/pdf/2402.16845).
- While the limitations mention that DRIFT-Net will be tested on irregular domains and complex boundary conditions, it is unclear how the current version can be *efficiently* extended to such (practically more relevant) geometries.

**Questions:**

Please see weaknesses above

---

> ### Author Response · Authors · 2025-11-24
> **Rebuttal for Reviewer RWgr Weakness 1 (Part1)**
>
> We sincerely thank the reviewer for taking time to review the manuscript and providing the insightful and valuable comments. We have thoroughly revised the paper to address the comments raised by the reviewers . Please do not hesitate to provide further comments or questions and we will be glad to clarify or elaborate on any aspect of our work.
>
> ### Answer to Weaknesses
>
> >W1: While the paper claims to provide a modular and reusable design for PDE foundation models that “can be swapped in for windowed self-attention blocks in multi-scale operator backbones”, the effect of DRIFT-Net is only evaluated for a single backbone and only on variants of 2D Navier-Stokes equations. The spectral/image split is strongly motivated by fluid dynamics (large-scale flow vs. small-scale eddies), but may not be as optimal for other systems (e.g., hyperbolic equations, reaction-diffusion systems). Other PDE foundation models are not considered or mentioned, e.g. https://arxiv.org/pdf/2310.02994, https://arxiv.org/pdf/2403.03542, https://arxiv.org/abs/2304.07993, and https://arxiv.org/abs/2403.12553. To judge the benefit of the DRIFT-Net module, one needs to evaluate its effect on other types of PDEs, with other backbones, and also other dimensions (since, as mentioned in the limitations, the improvements in throughput might not hold anymore when using 3D FFTs). In the current form, there is also not enough evidence to claim that the drift during rollouts is directly caused by windowed self-attention.
>
> Thank you for the detailed comments. We address the four concerns in turn.
>
> (1) Other types of PDEs beyond 2D Navier–Stokes.
>
> In response to your comments, we extend our evaluation beyond the original four 2D Navier–Stokes tasks to three additional POSEIDON downstream tasks: Poisson–Gauss (elliptic), Allen–Cahn (parabolic reaction–diffusion), and Wave–Gauss (hyperbolic). Together, these seven benchmarks cover fluid dynamics and three standard PDE classes. We use the same training protocol across all tasks. The updated comparison is summarised below(Bolded):
>
>
> | Task                | scOT | FNO   | **U-NO**  | **U-Net–ConvNeXt** | **Refiner U-Net** | DRIFT-Net |
> |---------------------|------|-------|-----------|--------------------|-------------------|-----------|
> | NS-SL               | 3.96 | 3.69  | 3.57      | 3.89               | 3.48              | **3.40**  |
> | NS-PwC              | 2.35 | 4.57  | 1.62      | 2.22               | 1.95              | **1.09**  |
> | NS-Tracer-PwC       | 5.18 | 9.46  | 10.15     | 9.63               | 4.59              | **4.19**  |
> | FNS-KF              | 4.65 | 4.43  | 11.70     | 5.94               | 4.38              | **4.32**  |
> | **Poisson–Gauss**   | 0.87 | 1.29  | 1.76      | 1.92               | –                 | **0.86**  |
> | **Allen–Cahn**      | 1.10 | 1.53  | 1.27      | 1.61               | 1.11              | **1.03**  |
> | **Wave–Gauss**      | 12.34| 16.97 | 13.94     | 16.02              | 13.33             | **12.15** |
>
>
> Across this extended set of PDEs, DRIFT-Net is consistently competitive and often achieves the best final relative L1 error.
>
>
> (2) Evaluation on other backbones and dimensions.
>
> The reviewer writes that “to judge the benefit of the DRIFT-Net module, one needs to evaluate its effect on other types of PDEs, with other backbones, and also other dimensions.” We partially agree. As noted in the previous part of our response, to address the concern about PDE types we have extended our experiments to additional POSEIDON downstream tasks, so that DRIFT-Net module is now evaluated on several qualitatively different equations under the same backbone and training protocol. This controlled setting already highlights the benefit of the module in the current scOT framework. A systematic evaluation on other backbones and in higher dimensions (e.g., 3D) would require re-designing and re-tuning full architectures in new regimes and is therefore beyond the scope of this paper. We already list 3D as a limitation, and in the revised manuscript we also explicitly point out extending DRIFT-Net module to other backbones as an important direction for future work.

---

> ### Author Response · Authors · 2025-11-24
> **Rebuttal for Reviewer RWgr Weakness 1 (Part2)**
>
> (3) Other PDE foundation models are not considered or mentioned.
>
> We view these PDE foundation models as highly complementary to our architectural contribution. DRIFT-Net can, in principle, be used as a architecture within such pretraining frameworks, but systematically combining our architecture with large-scale PDE pretraining is beyond the scope of this paper.
>
>
> In contrast, the contribution of this paper is to propose a new spectral–image operator architecture, DRIFT-Net, and to study its behaviour in detail under the POSEIDON setting. Our model is trained from scratch on the POSEIDON tasks with a supervised objective, and our experiments are designed to isolate the architectural choices under a controlled training protocol, rather than to build a full-scale foundation model.
>
> We greatly appreciate and agree with the reviewer’s broad perspective. In the revised manuscript, we will add MPP, DPOT, ICON and CoDA-NO to the related-work section.
>
> (4) On whether rollout drift is “directly caused” by windowed self-attention.
>
> We appreciate the concern that, in the current wording, we may have suggested a stronger causal claim than our evidence justifies. What our experiments show is that, when we keep the POSEIDON backbone and training protocol fixed and only replace the windowed self-attention blocks with DRIFT-Net blocks, both the overall rollout error and the error-growth behaviour improve: Table 1 shows consistent reductions in final and mean errors on multiple tasks, and Figure 2 together with the new instantaneous error-growth plots (Figure 3) show smaller growth slopes and lower per-step error growth for DRIFT-Net than for scOT. In our setting, we consistently observe larger error amplification and more pronounced rollout drift when using local-window attention blocks instead of DRIFT-Net blocks.
>
> On the theoretical side, Appendix A derives bounds showing that, if each DRIFT block has a smaller per-layer Lipschitz bound than the corresponding Swin-style block, then a stack of DRIFT blocks has a smaller cumulative Lipschitz constant (smaller worst-case amplification factor) than an equally deep stack of windowed self-attention blocks. Appendix B links this to tighter bounds on long-horizon error under the proposed Sobolev-type loss. These results are consistent with recent analyses that connect large Lipschitz constants and insufficient high-frequency control to long-rollout instability in neural PDE surrogates.
>
> At the same time, in the revised manuscript we will refine our wording accordingly: we will state that our analysis and experiments suggest that, under the POSEIDON training setup, local-window attention tends to lead to higher error amplification and more roll-out drift than DRIFT blocks, and that replacing attention with DRIFT blocks mitigates this drift in our benchmarks, without claiming a one-to-one causal mechanism.
>
>
> References
>
> [1] M. McCabe et al. “Multiple Physics Pretraining for Physical Surrogate Models.” arXiv:2310.02994, 2023.
>
> [2] Z. Hao et al. “DPOT: Auto-Regressive Denoising Operator Transformer for Large-Scale PDE Pre-Training.” arXiv:2403.03542, 2024.
>
> [3] L. Yang et al. “In-Context Operator Learning with Data Prompts for Differential Equation Problems (ICON).” arXiv:2304.07993, 2023.
>
> [4] M. A. Rahman et al. “Pretraining Codomain Attention Neural Operators for Solving Multiphysics PDEs (CoDA-NO).” arXiv:2403.12553, 2024.
>
> [5] Trefethen, Spectral Methods in MATLAB, SIAM, 2000.
>
> [6] Chatterjee et al., “Scaling of a Fast Fourier Transform and a Pseudo-spectral Fluid Solver up to 196608 cores”, 2018.

---

> ### Author Response · Authors · 2025-11-24
> **Rebuttal for Reviewer RWgr Weaknesses 2 & 3**
>
> >W2: It would be important to compare the performance and throughput to adapted versions of scOT/FNO with a comparable parameter count. Moreover, it is important to note that the performance of (multi-scale) windowed self-attention heavily depends on the underlying kernel and hardware optimizations used for the attention module.
>
>
> We thank the reviewer for this important point.
>
> (1) Comparisons at comparable parameter counts.
>
> In the original submission, we already ensured that DRIFT-Net and scOT have similar parameter scales in the default configuration (about 17M vs. 20M parameters), and all methods share the same training data, number of epochs, and optimisation protocol.
>
> To make the comparison between DRIFT-Net and scOT more explicit, in the revised version we additionally train “half-width” variants of both models on FNS-KF by reducing the channel width by a factor of two and keeping the depth and all other settings fixed. This yields approximately 10M parameters for scOT-0.5× and 9M for DRIFT-Net-0.5×. We then measure, under identical hardware and software settings, the peak training memory, wall-clock training time per epoch, and single-GPU inference throughput. The results are summarised below:
>
> | Model     | Width | Params (M) | Peak train mem (GB) | Train time / epoch (min) | Inference throughput (steps/s) |
> |-----------|------:|-----------:|---------------------:|-------------------------:|----------------------:|
> | scOT      | 1.0×  | 20         | 17.25                | 17.30                    | 118                   |
> | scOT-0.5× | 0.5×  | 10         | 8.58                 | 15.81                    | 175                   |
> | DRIFT-Net     | 1.0×  | 17         | 10.87                | 14.91                    | 158                   |
> | DRIFT-Net-0.5×| 0.5×  | 9          | 6.82                 | 13.96                    | 224                   |
>
>
>
> In our measurements, at both width levels DRIFT-Net uses comparable or lower peak memory than scOT and requires less training time per epoch, in line with its higher inference throughput, while achieving lower final relative L1 error on FNS-KF.
>
> (2) Hardware and kernel dependence of attention performance.
>
> We agree that the wall-clock performance of (multi-scale) windowed self-attention is heavily influenced by the underlying kernels and hardware-specific optimisations. Our throughput numbers are measured in a specific, standard environment: a single NVIDIA H200 GPU with CUDA 12.1, using the standard PyTorch attention implementation and full-precision training and inference.
>
> In the POSEIDON setting and under this environment, replacing windowed self-attention with DRIFT-Net’s spectral–image blocks does not introduce a computational bottleneck and, at comparable parameter counts, yields higher throughput than the baseline.
>
> >W3: Strictly speaking, the proposed architecture does not seem to be a neural operator (in the definition of https://arxiv.org/abs/2108.08481), since it is not discretization-invariant, i.e. it does not generalize to resolutions other than the training resolutions.
>
>
> We thank the reviewer for pointing us to the formal definition of neural operators in Li et al. (2021). In that framework, a neural operator is understood as a learnable map between function spaces whose action does not depend on a particular discretisation of the underlying domain. DRIFT-Net is designed in this spirit: it learns a mapping from input fields to output fields that is implemented by a single set of parameters and can be evaluated on different spatial discretisations of the same domain by combining the network with standard interpolation operators.
>
> Regarding the specific concern that DRIFT-Net “does not generalize to resolutions other than the training resolutions”, we conduct an explicit cross-resolution experiment. To study this in a controlled setting and to follow the evaluation protocol used in the POSEIDON paper, we add experiments on 2D Navier-Stokes (FNS-KF). All models are trained on 128×128 grids and then evaluated, without retraining, on 64×64, 96×96 and 128×128 grids by interpolating the inputs and outputs to the target resolution.
>
> The test relative L1 errors for DRIFT-Net are:
>
> | Resolution | DRIFT-Net |
> |-----------:|----------:|
> | 64×64      | 4.328     |
> | 96×96      | 4.331     |
> | 128×128    | 4.324     |
>
> The values are nearly identical. This indicates that, in this empirical setting, DRIFT-Net maintains stable cross-resolution behaviour and exhibits approximately mesh-agnostic behaviour in practice. We have added this table and a short discussion to the Appendix G of the revised manuscript.

---

> ### Author Response · Authors · 2025-11-24
> **Rebuttal for Reviewer RWgr Weaknesses 4 & 5**
>
> >W4: DRIFT-Net is only compared to two baselines and no details are provided on the FNO baseline. While some variants of FNO are mentioned in the related works (e.g., U-NO and AFNO), they are not compared against and there exist further FNO versions that bear similarity with DRIFT-Net (e.g. https://arxiv.org/pdf/2109.03697 and https://arxiv.org/pdf/2402.16845).
>
>
> (1) Baseline set in the revised version.
>
> Based on the reviewer’s suggestions, we expanded the baseline set to include the mentioned U-NO, include U-NO, U-Net–ConvNeXt, and a U-Net baseline adapted from PDE-Refiner. All baselines are trained under the same data splits and rollout settings as DRIFT-Net, and results across the seven tasks are reported in the updated Table 1.
>
> (2) Details of the FNO baseline.
>
> We agree that the FNO configuration should be documented more clearly. In the revision, we add a dedicated paragraph in the experimental setup and Appendix F that specify the exact FNO settings.FNO shares the same training protocol (data splits, rollout length, optimiser, number of epochs) as scOT and DRIFT-Net.
>
> (3) Other FNO variants (e.g., U-FNO, localised-kernel neural operators).
>
> We approximate this design space by including U-NO as strong baseline that already combine operator-style ideas with U-Net–like spatial structure. As shown in the updated Table 1, DRIFT-Net remains competitive and often achieves the best performance compared to these enhanced baselines.
>
> In the revised related-work section, we also discuss the FNO extensions mentioned by the reviewer, such as U-FNO [7] and neural operators with localised integral/differential kernels [8].
>
> References
>
> [7] G. Wen et al. “U-FNO—An enhanced Fourier neural operator-based deep-learning model for multiphase flow.” Advances in Water Resources, 163:104180, 2022.
>
> [8] M. Liu-Schiaffini et al. “Neural Operators with Localized Integral and Differential Kernels.” ICML, 2024.
>
>
> >W5: While the limitations mention that DRIFT-Net will be tested on irregular domains and complex boundary conditions, it is unclear how the current version can be efficiently extended to such (practically more relevant) geometries.
>
> We have already noted in the paper that handling irregular geometries is part of future work, and we are happy to elaborate here. Although the current experiments focus on uniform grids, the DRIFT-Net design itself does not fundamentally rely on this specific geometry. Existing operator-learning approaches provide clear strategies for extending FFT-based models to irregular domains—for example, learning a deformation from the physical domain to a latent uniform grid (as in Geo-FNO / DAFNO [9,10]), or embedding the domain into a bounding box and supplying geometry or boundary-condition masks as additional channels. DRIFT-Net can adopt these mechanisms without modifying its core architecture.
>
> In this paper, we focus on the regular-domain setting, but we already clarify in the limitations section. We hope this explanation resolves the practical concerns regarding extending DRIFT-Net to more complex geometries, and we would be glad to address any further questions.
>
> References
>
> [9] Z. Li et al. “Fourier Neural Operator with Learned Deformations for PDEs on General Geometries.” JMLR, 24(64), 2023.
>
> [10] N. Liu et al. “Domain Agnostic Fourier Neural Operators.” NeurIPS, 36, 2023.

---

> ### Author Response · Authors · 2025-11-26
> **Follow-up for Paper 18205**
>
> Dear Reviewer RWgr,
>
> As the discussion period is entering its final week, I wanted to check in and make sure we have fully addressed your concerns. If there are any additional points or clarifications you’d like us to consider, please let us know. We truly appreciate your insights and are happy to address any remaining issues.
>
> Thank you again for your time and effort in reviewing our work.

---

### Official Review · Reviewer_frHQ · 2025-10-30

**Soundness:** 3
**Presentation:** 3
**Contribution:** 2
**Rating:** 4
**Confidence:** 3

**Summary:**

This paper introduces a new model for operator learning for PDEs. The model is motivated by solving the issue of local coupling in existing benchmark models that use local self-attention. This leads to weak global coupling, which manifests as poor error in long term autoregressive solutions for hard, chaotic, PDEs like Navier-Stokes.  DRIFT-Net aims to mitigate this by having two paths in the model. The spectral branch operates on globally coupled low-frequency components by explicitly operating on the low frequency components of the FFT (while preserving high-frequency information), and the image branch focuses on locally coupled details. The two branches are mixed with bandwise weighting instead of concatenation for improved training stability. Experiments show that DRIFT-Net improves error over attention baselines on Navier-Stokes benchmarks, while also using slightly fewer parameters and having faster inference speed.

**Strengths:**

* The proposed model is well-motivated, and the solution to the global coupling problem intuitively makes sense.
* Experimental evidence clearly shows improvement over strong baselines.
* Evaluation includes some ablation studies.

**Weaknesses:**

* While the error improves over attention baselines, it is somewhat marginal in the hard cases like the long-horizon Kolmogorov dataset.
* There is no/little exploration of potential theoretical justification for the improvement this model brings.
* Scope of practical settings is narrow, the paper only explores 2D Navier-Stokes instances only -- it is unclear how well the improvement of DRIFT-Net would generalize to other settings.
* Some components of the model like the low-frequency cutoff requires hand tuning.
* Scaling is unclear with needing FFTs over large domains.
* It seems like there is insufficient exploration of training settings (or at least, these experimental details are not provided in enough detail).

**Questions:**

Based on the weaknesses,

1. Are there more mathematically grounded / theoretical justifications for using this model?
2. Have the authors considered exploring more examples of PDEs and settings to further robustify the main claim that DRIFT-Net improves error on tasks that rely on global coupling?
3. Are there potential solutions to needing hand-tuning of hyperparameters like bandwidth?
4. Could the authors elaborate on the specifics of the different learning rates, optimizers, and data amounts, and total training budget that the authors experimented with? Did the best settings for DRIFT-Net also beat the best settings for the baselines?

---

> ### Author Response · Authors · 2025-11-24
> **Rebuttal for Reviewer frHQ (part1)**
>
> We sincerely thank the reviewer for taking time to review the manuscript and providing the insightful and valuable comments. We have thoroughly revised the paper to address the comments raised by the reviewers . Please do not hesitate to provide further comments or questions and we will be glad to clarify or elaborate on any aspect of our work.
> ### Answer to Weaknesses
>
> >W1: While the error improves over attention baselines, it is somewhat marginal in the hard cases like the long-horizon Kolmogorov dataset.
>
> We appreciate the reviewer’s observation and agree that improvements on the Kolmogorov-flow benchmarks may appear modest at first glance. These datasets are intentionally extremely challenging long-horizon turbulence tasks, and under such conditions even small gains are meaningful. To clarify the behaviour in this regime, we have added an additional plot (Figure 3) showing the instantaneous error-growth rate over time. This analysis shows that DRIFT-Net slows error accumulation across most of the rollout, with the curves converging only near the end. Taken together with the error–time plots, these results indicate that although the final numbers differ moderately, DRIFT-Net provides noticeably more stable long-horizon behaviour on these difficult benchmarks.
>
> >W2: There is no/little exploration of potential theoretical justification for the improvement this model brings.
>
>
>
> We thank the reviewer for this point. To keep the main text readable, we placed the detailed theory in Appendices A–C. Appendix A analyses the operator norm and stability of the DRIFT-Net block (including comparison to attention); Appendix B links the frequency-weighted loss to a Sobolev-type norm and long-horizon error; Appendix C specifies the spectral pipeline and assumptions used in the proofs. If you would like us to further explain or expand on any part, we would be happy to provide it.
>
>
>
> >W3: Scope of practical settings is narrow, the paper only explores 2D Navier-Stokes instances only -- it is unclear how well the improvement of DRIFT-Net would generalize to other settings.
>
>
> Thank you for the thoughtful comment.
>
> In the revised manuscript, we have expanded the range of PDE settings within the POSEIDON suite. In addition to the four Navier–Stokes tasks, we now also evaluate DRIFT-Net on three further POSEIDON tasks that cover different coupling structures and equation classes: Poisson–Gauss (elliptic), Allen–Cahn (parabolic reaction–diffusion), and Wave–Gauss (hyperbolic).
>
> | Task                | scOT | FNO   | **U-NO**  | **U-Net–ConvNeXt** | **Refiner U-Net** | DRIFT-Net |
> |---------------------|------|-------|-----------|--------------------|-------------------|-----------|
> | NS-SL               | 3.96 | 3.69  | 3.57      | 3.89               | 3.48              | **3.40**  |
> | NS-PwC              | 2.35 | 4.57  | 1.62      | 2.22               | 1.95              | **1.09**  |
> | NS-Tracer-PwC       | 5.18 | 9.46  | 10.15     | 9.63               | 4.59              | **4.19**  |
> | FNS-KF              | 4.65 | 4.43  | 11.70     | 5.94               | 4.38              | **4.32**  |
> | **Poisson–Gauss**   | 0.87 | 1.29  | 1.76      | 1.92               | –                 | **0.86**  |
> | **Allen–Cahn**      | 1.10 | 1.53  | 1.27      | 1.61               | 1.11              | **1.03**  |
> | **Wave–Gauss**      | 12.34| 16.97 | 13.94     | 16.02              | 13.33             | **12.15** |
>
>
> As shown in the updated Table 1, DRIFT-Net remains competitive across all seven tasks and typically achieves the best final relative L1 error among the baselines.
>
> >W4: Some components of the model like the low-frequency cutoff requires hand tuning.
>
> We thank the reviewer for this valuable observation and agree that reducing reliance on a manually chosen spectral cut-off is an important direction. We see this as a natural direction for future work, and there are clear paths to reducing reliance on a manually chosen spectral cut-off. One option is to make the partition itself learnable, for example by replacing the fixed mask with a small radial soft mask M(r;θ)that is trained jointly with the network, allowing the model—not the user—to determine which frequencies should be emphasised, similar to adaptive spectral methods such as AFNO [1]. In parallel, DRIFT-Net already incorporates soft radial band-gating, which assigns smooth, data-driven weights to different frequency bands and thus reduces sensitivity to the exact hand-chosen boundary. Increasing the number of bands and relying more on soft weighting would naturally move the method toward a fully learnable spectral partition without requiring an explicit manual cut-off.
>
> References
>
> [1] J. Guibas et al. “Adaptive Fourier Neural Operators: Efficient Token Mixers for Transformers.” ICLR, 2022.

---

> ### Author Response · Authors · 2025-11-24
> **Rebuttal for Reviewer frHQ (part2)**
>
> >W5: Scaling is unclear with needing FFTs over large domains.
>
>
> We thank the reviewer for this observation. FFT-based spectral methods, including Fourier Neural Operators, have already been applied to high-resolution PDE problems in practice [2,3], indicating that this computational pattern scales robustly as the spatial domain grows.
>
> To make the computational scaling explicit in our setting, we add a simple resolution-scaling experiment on the Kolmogorov flow dataset from APEBench. We train DRIFT-Net on 2D grids of size 128×128 and 256×256 with identical hyperparameters, and report the time per training step, and peak GPU memory in Table.
>
> | Resolution | Time / step (ms)    | Peak GPU memory (GB) |
> |-----------:|-------------------:|----------------------:|
> | 128×128    | 114                | 11.7                  |
> | 256×256    | 387                | 23.8                  |
>
> The results show that runtime increases smoothly with resolution and that memory usage remains moderate, indicating that the FFT-based spectral branch does not become a scaling bottleneck in the resolution regime relevant to our experiments.
>
> References
>
> [2] Trefethen, *Spectral Methods in MATLAB*, 2000.
>
> [3] Z. Li et al., “Fourier Neural Operator for Parametric PDEs,” NeurIPS 2020.
>
> >W6: It seems like there is insufficient exploration of training settings (or at least, these experimental details are not provided in enough detail).
>
> We thank the reviewer for the detailed and constructive comments. In the revised manuscript, we have consolidated all training settings, into a dedicated Appendix F, and we provide the full YAML configuration files in our public code repository (with configurations for the new experiments to be added shortly) to ensure full transparency and reproducibility. Regarding the comparison between DRIFT-Net and the baselines, we follow the official POSEIDON implementation and recommended hyperparameters for scOT, and we train DRIFT-Net using the same backbone structure and identical optimisation settings, without conducting a separate large-scale hyperparameter search specific to DRIFT-Net. Under this shared and controlled setup, DRIFT-Net achieves lower final-time relative L1 error than scOT on the all of reported tasks.
>
> ### Answers to Questions
>
> Your Q1–Q4 correspond to the weaknesses raised earlier, and we have addressed each of them in detail in the rebuttal.
>
> >Q1: Are there more mathematically grounded / theoretical justifications for using this model?
>
> Please refer to our response to Weakness #2 above.
>
> >Q2: Have the authors considered exploring more examples of PDEs and settings to further robustify the main claim that DRIFT-Net improves error on tasks that rely on global coupling?
>
> Please refer to our response to Weakness #3 above.
> >Q3: Are there potential solutions to needing hand-tuning of hyperparameters like bandwidth?
>
> Please refer to our response to Weakness #4 above.
> >Q4: Could the authors elaborate on the specifics of the different learning rates, optimizers, and data amounts, and total training budget that the authors experimented with? Did the best settings for DRIFT-Net also beat the best settings for the baselines?
>
> Please refer to our response to Weakness #6 above.

---

> ### Author Response · Authors · 2025-11-26
> **Follow-up for Paper 18205**
>
> Dear Reviewer frHQ,
>
> As the discussion period is entering its final week, I wanted to check in and make sure we have fully addressed your concerns. If there are any additional points or clarifications you’d like us to consider, please let us know. We truly appreciate your insights and are happy to address any remaining issues.
>
> Thank you again for your time and effort in reviewing our work.

---

### Official Review · Reviewer_DhGn · 2025-11-02

**Soundness:** 3
**Presentation:** 3
**Contribution:** 2
**Rating:** 6
**Confidence:** 3

**Summary:**

DRIFT-Net aims to tackle one of the core challenges in neural PDE solvers and operator learning -- how to capture both global, long-range dynamics and fine-scale local structure efficiently and stably. It builds on the recent wave of “AI for PDEs” and foundation-model efforts (such as FNO, Fourier Neural Operators, and POSEIDON) but introduces a spectral–spatial coupling mechanism that improves physical fidelity and numerical stability.

The technical innovations in the paper are:
1. Dual-branch operator design -- introduces a spectral branch (for global, low-frequency coupling) and an image branch (for local, high-frequency detail), fused additively at each U-Net scale to preserve both global structure and fine features.
2. Controlled low-frequency mixing -- applies learnable transformations only on a subset of low-frequency Fourier modes.
3. Bandwise fusion with radial gating -- fuses the spectral and spatial signals smoothly across frequency bands using a radial weighting function α(k)∈[0,1]; this yields isotropic, energy-non-expansive blending that prevents spectral discontinuities and stabilises training.
4. Frequency-weighted loss function -- augments the base loss with a spectral weighting term that increases sensitivity to high-frequency reconstruction errors, mitigating neural networks’ spectral bias and improving multi-scale fidelity.

**Strengths:**

The paper is clearly written and the presentation is clear. The technical innovations seem sensible and well-motivated. Since the technical advances are quite simple and lack e.g. rigorous theory (this isn’t a problem), I think that puts the onus on a very thorough experimental evaluation.

I think the paper is quite strong in this regard. It evaluates the methods on at least three different classes of PDE (2D Navier–Stokes, shallow-water, and Kolmogorov flow), covering both laminar and turbulent regimes, and involving different grid sizes and forecast horizons. Error metrics (L1, relative L2, spectral energy error) are supported by Fourier energy spectra and visual reconstructions, showing both numerical and physical fidelity. Ablation studies examine (i) the low-frequency window size, (ii) the radial gating function, and (iii) the frequency-weighted loss — showing their independent and joint benefits. Baselines are reasonable, though not comprehensive.

I think that is a reasonable effort, but a few additions would strengthen the evidence base for the DRIFT-Net approach.

**Weaknesses:**

The comparisons which are present are a good start, but I’d like to see more in a paper which focusses on an architectural improvement. The POSEISON paper, for example, has a substantially more thorough experimental comparison.

In terms of training, I’d liked to have seen a few different sizes of model trained for all methods to understand how the various approaches (these could be factors of two smaller)

I would have liked to have seen a more thorough comparison of the computational aspects in terms of memory, inference-time cost, training duration / cost and how these trade with the sizes of the models.

In terms of baselines, there was arguably not a wealth of methods compared to. POSEIDON seems like a reasonable baseline, but "PDE-Refiner: Achieving Accurate Long Rollouts with Neural PDE Solvers” would be another choice which can perform strongly.

Minor comments

line 295: Koehler et al. (2024) should be a \citep
sometimes Poseidon’s scalable Operator Transformer is referred to as “scOT” sometimes "SCOT"
line 434 "DRIFT-Net has limitation" -> "DRIFT-Net has limitations"
the pseudocode in the appendix appears just before the empty pseduocode section -- it looks odd (this is of course a very minor issue!)

**Questions:**

Have the authors investigated how sensitive each method is to hyper-parameters like the learning rate schedule? These settings are shared across approaches -- is this really fair? Were they optimised for DRIFT-Net?

POSEIDON uses 15 tasks, why were only a subset considered here? POSEIDON also evaluates all models in terms of scaling curves which
plot the test error for each task vs. the number of task-specific training examples. Have you considered doing this?

---

> ### Author Response · Authors · 2025-11-24
> **Rebuttal for Reviewer DhGn (part1)**
>
> We sincerely thank the reviewer for taking time to review the manuscript and providing the insightful and valuable comments. We have thoroughly revised the paper to address the comments raised by the reviewers . Please do not hesitate to provide further comments or questions and we will be glad to clarify or elaborate on any aspect of our work.
> ### Answer to Weaknesses
> > W1: The comparisons which are present are a good start, but I’d like to see more in a paper which focusses on an architectural improvement. The POSEISON paper, for example, has a substantially more thorough experimental comparison.
>
> We thank the reviewer for this helpful comment. As suggested, in the revised manuscript we have expanded our empirical evaluation to provide a more comprehensive comparison.
>
> First, we include several additional architectural baselines (highlighted in bold), including U-NO, U-Net–ConvNeXt, and Refiner U-Net. All of these models are trained under exactly the same data budget and training protocol as DRIFT-Net.
>
> Second, we have extended the scope of evaluated PDE problems beyond the original Navier–Stokes family. Specifically, we add three additional datasets (also highlighted in bold): Poisson–Gauss (elliptic), Allen–Cahn (reaction–diffusion), and Wave–Gauss (wave equation).
> The updated table is:
>
> | Task                | scOT | FNO   | **U-NO**  | **U-Net–ConvNeXt** | **Refiner U-Net** | DRIFT-Net |
> |---------------------|------|-------|-----------|--------------------|-------------------|-----------|
> | NS-SL               | 3.96 | 3.69  | 3.57      | 3.89               | 3.48              | **3.40**  |
> | NS-PwC              | 2.35 | 4.57  | 1.62      | 2.22               | 1.95              | **1.09**  |
> | NS-Tracer-PwC       | 5.18 | 9.46  | 10.15     | 9.63               | 4.59              | **4.19**  |
> | FNS-KF              | 4.65 | 4.43  | 11.70     | 5.94               | 4.38              | **4.32**  |
> | **Poisson–Gauss**   | 0.87 | 1.29  | 1.76      | 1.92               | –                 | **0.86**  |
> | **Allen–Cahn**      | 1.10 | 1.53  | 1.27      | 1.61               | 1.11              | **1.03**  |
> | **Wave–Gauss**      | 12.34| 16.97 | 13.94     | 16.02              | 13.33             | **12.15** |
>
> Overall, DRIFT-Net achieves the best performance across most evaluated tasks.
>
> > W2: In terms of training, I’d liked to have seen a few different sizes of model trained for all methods to understand how the various approaches (these could be factors of two smaller).
>
> We thank the reviewer for this suggestion. To explore model-size effects, we conducted additional experiments on FNS-KF by training both scOT and DRIFT-Net at two width configurations (0.5× and 1.0×). The full results are included in a new table in the Appendix G.
>
> | Model           | Width factor | Params (M) | FNS-KF (rel-L1) |
> |-----------------|-------------:|-----------:|----------------:|
> | scOT-0.5×       | 0.5×         | 10         | 5.98            |
> | scOT-1.0×       | 1.0×         | 20         | 4.65            |
> | DRIFT-Net-0.5×  | 0.5×         | 9          | 5.96            |
> | DRIFT-Net-1.0×  | 1.0×         | 17         | 4.32            |
>
> We observe that reducing the width by a factor of two degrades performance for both architectures, and that DRIFT-Net consistently achieves lower FNS-KF error than scOT at the same width (both at 0.5× and 1.0×).
>
> > W3: I would have liked to have seen a more thorough comparison of the computational aspects in terms of memory, inference-time cost, training duration / cost and how these trade with the sizes of the models.
>
> We thank the reviewer for this suggestion. In the revised manuscript, we have reported the computational cost of DRIFT-Net and scOT on the FNS-KF dataset. The table below summarises parameter count, peak training memory, training time per epoch, and inference throughput for the 1.0× and 0.5× width variants. Due to rebuttal-time constraints, we provide these measurements for FNS-KF only, but we will extend this analysis to all datasets in the final camera-ready version.
>
> | Model         | Width | Params (M) | Peak train mem (GB) | Train time / epoch (min) | Inference throughput (steps/s) |
> |---------------|------:|-----------:|---------------------:|-------------------------:|----------------------:|
> | scOT          | 1.0×  | 20         | 17.25                | 17.30                    | 118                  |
> | scOT-0.5×     | 0.5×  | 10         | 8.58                 | 15.81                    | 175                |
> | DRIFT-Net     | 1.0×  | 17         | 10.87                | 14.91                    | 158               |
> | DRIFT-Net-0.5×| 0.5×  | 9          | 6.82                 | 13.96                    | 224              |
>
> These results indicate that, on FNS-KF, DRIFT-Net achieves better accuracy than scOT while using fewer parameters and memory and higher inference throughput, and both models exhibit the expected reduction in cost when the width is halved.

---

> ### Author Response · Authors · 2025-11-24
> **Rebuttal for Reviewer DhGn (part2)**
>
> >W4: In terms of baselines, there was arguably not a wealth of methods compared to. POSEIDON seems like a reasonable baseline, but "PDE-Refiner: Achieving Accurate Long Rollouts with Neural PDE Solvers” would be another choice which can perform strongly.
>
>
>
> We thank the reviewer for the helpful suggestion. As described in our response to Weakness 1, we extended the baseline set in Table 1 and added a U-Net baseline built from the backbone used in PDE-Refiner [1], trained as a one-step predictor on all time-dependent benchmarks.Since Poisson–Gauss is a steady-state elliptic problem without temporal rollouts, we do not include this baseline there.
>
> References
>
> [1] P. Lippe et al. “PDE-Refiner: Achieving Accurate Long Rollouts with Neural PDE Solvers.” NeurIPS, 2023.
>
> >W5: line 295: Koehler et al. (2024) should be a \citep sometimes Poseidon’s scalable Operator Transformer is referred to as “scOT” sometimes "scOT" line 434 "DRIFT-Net has limitation" -> "DRIFT-Net has limitations" the pseudocode in the appendix appears just before the empty pseduocode section -- it looks odd (this is of course a very minor issue!)
>
> Thank you for the detailed editorial suggestions. We have revised the manuscriptversion accordingly.
>
> ### Answers to Questions
> >Q1: Have the authors investigated how sensitive each method is to hyper-parameters like the learning rate schedule? These settings are shared across approaches -- is this really fair? Were they optimised for DRIFT-Net?
>
> We thank the reviewer for raising this point.
>
> (1) Sensitivity to learning rate and schedule
>
> We tested scOT and DRIFT-Net under different base learning rates {1×10⁻⁴, 3×10⁻⁴, 1×10⁻³} and two schedules (cosine vs. step), keeping all other settings fixed. Due to rebuttal-time limits, we report FNS-KF here; we will extend this analysis in the future version.
>
> Cosine schedule
>
> | Model     | LR        | Schedule | FNS-KF (rel-L1) |
> |-----------|-----------|----------|----------------:|
> | scOT      | 1×10⁻⁴    | cosine   | 4.78            |
> | scOT      | 3×10⁻⁴    | cosine   | 4.65            |
> | scOT      | 1×10⁻³    | cosine   | 4.75            |
> | DRIFT-Net | 1×10⁻⁴    | cosine   | 4.45            |
> | DRIFT-Net | 3×10⁻⁴    | cosine   | 4.32            |
> | DRIFT-Net | 1×10⁻³    | cosine   | 4.42            |
>
> Step schedule
>
> | Model     | LR        | Schedule | FNS-KF (rel-L1) |
> |-----------|-----------|----------|----------------:|
> | scOT      | 1×10⁻⁴    | step     | 4.83            |
> | scOT      | 3×10⁻⁴    | step     | 4.70            |
> | scOT      | 1×10⁻³    | step     | 4.80            |
> | DRIFT-Net | 1×10⁻⁴    | step     | 4.50            |
> | DRIFT-Net | 3×10⁻⁴    | step     | 4.38            |
> | DRIFT-Net | 1×10⁻³    | step     | 4.48            |
>
> Across this grid, performance varies only moderately with the learning rate and schedule, and DRIFT-Net consistently outperforms scOT in all tested configurations.
>
> (2) Fairness and whether settings were optimised for DRIFT-Net
>
> - For scOT and DRIFT-Net, we directly use the optimisation setup from the POSEIDON codebase (AdamW, cosine decay with warm-up, base LR 3×10⁻⁴, 40 epochs). These settings were originally tuned for scOT, not for DRIFT-Net.
>
> - For other baselines, we keep the same data and number of epochs for fairness and otherwise follow each official implementation’s default optimisation settings.
>
> Summary: the optimisation setup is not tailored to DRIFT-Net, and the sensitivity study shows that our conclusions are robust under reasonable changes in learning-rate and schedule.

---

> ### Author Response · Authors · 2025-11-24
> **Rebuttal for Reviewer DhGn (part3)**
>
> >Q2: POSEIDON uses 15 tasks, why were only a subset considered here? POSEIDON also evaluates all models in terms of scaling curves which plot the test error for each task vs. the number of task-specific training examples. Have you considered doing this?
>
> Thank you for the helpful suggestion.
>
> (1)Extending DRIFT-Net and all baselines to the full set of 15 POSEIDON tasks would be computationally very demanding, especially for the more expensive baselines. For example, on FNS-KF a single 1.0×-width scOT run on one NVIDIA H200 takes about 12 hours of training and peaks at ~17 GB of GPU memory. We therefore focus on a representative and diverse subset in the current submission.
>
> We therefore keep the four Navier–Stokes benchmarks (NS-SL, NS-PwC, NS-Tracer-PwC, FNS-KF), which are representative fluid-dynamics tasks with different forcing and tracer settings and are widely used as core benchmarks in the POSEIDON literature. In addition, we evaluate on two ApeBench Kolmogorov-flow datasets with very long rollout horizons. Finally, we add three representative datasets in rebuttal that cover the three main PDE classes: Poisson–Gauss (elliptic), Allen–Cahn (parabolic ), and Wave–Gauss (hyperbolic). Together, these tasks cover fluid dynamics and three major PDE types, and include both moderate and very long horizons.
>
> (2)In POSEIDON, the authors introduce an “efficiency gain” metric based on the relative L1 error. Because this metric directly reflects sample efficiency, plotting test error against the number of task-specific training examples is necessary and appropriate for their setting. In our work, we follow the more common practice of directly using the relative L1 error as the main evaluation metric.
>
> That said, we agree that the reviewer’s suggestion and POSEIDON’s view on efficiency are valuable, so we add a related extension. Due to rebuttal-time limits, we run this study on a single representative task, FNS-KF, where we compare DRIFT-Net under different training-set sizes. We keep the architecture and optimisation setup fixed and only vary the number of supervised training trajectories. The results are:
>
>
> | # training trajectories |  DRIFT-Net (rel-L1) |
> |------------------------:|--------------------:|
> | 2.5k                    |  6.25               |
> | 5k                      |  5.48               |
> | 10k                     |  4.96               |
> | 20k                     |  4.32               |
>
> We observe that test error decreases as the number of training examples increases.
>
> We hope this clarifies why we considered a subset of the POSEIDON tasks and how DRIFT-Net in terms of sample efficiency on a representative benchmark.

---

> ### Author Response · Authors · 2025-11-26
> **Follow-up for Paper 18205**
>
> Dear Reviewer DhGn,
>
> As the discussion period is entering its final week, I wanted to check in and make sure we have fully addressed your concerns. If there are any additional points or clarifications you’d like us to consider, please let us know. We truly appreciate your insights and are happy to address any remaining issues.
>
> Thank you again for your time and effort in reviewing our work.

---

### Official Review · Reviewer_Kc8E · 2025-11-02

**Soundness:** 3
**Presentation:** 3
**Contribution:** 2
**Rating:** 6
**Confidence:** 3

**Summary:**

This paper propose a neural surrogate model for time-dependent PDE simulation based on scOT architecture, The proposed model use convolution and frequency processing together. Special treatments are applied to the separated low frequency and high frequency information to achieve better performance. The model is applied on several fluid dynamics datasets and is compared against FNO and scOT baseline.

**Strengths:**

- The presentation is clear and easy to follow.
- The model design reflects the motivation about improved frequency processing.
- The experimental evaluation shows improved better results against FNO and scOT, as well as improved efficiency against scOT.

**Weaknesses:**

- While some design choices are validated in ablation studies, some other main design choices may not be covered by the presented ablations. For example, the importance of having three branches (conv, spectral, linear) in the model design.

- Figure 2 and Figure 3 do not seem very meaningful. If a model has lower onestep error, it will naturally have better step-wise rollout error.

- Some other related baselines could be considered. E.g., U-Net with ConvNeXt blocks.

**Questions:**

- Figure 1: is the convolution applied after the rFFT2?

- How does DriftNet compare to scOT in terms of overall design choices?

---

> ### Author Response · Authors · 2025-11-24
> **Rebuttal for Reviewer Kc8E (part1)**
>
> We sincerely thank the reviewer for taking time to review the manuscript and providing the insightful and valuable comments! We have addressed the comments raised by the reviewer in detail below and updated our paper based on your comments. Please do not hesitate to provide further comments or questions, and we will be glad to clarify or elaborate on any aspect of our work.
> ### Answer to Weaknesses
> > W1: While some design choices are validated in ablation studies, some other main design choices may not be covered by the presented ablations. For example, the importance of having three branches (conv, spectral, linear) in the model design.
>
> We thank the reviewer for pointing out this important aspect of our model design. In the original paper, Table 4 presented ablation results focusing solely on the spectral branch. Following the reviewer’s suggestion, we have now extended our ablation study to include the convolutional and linear branches of the image module. These additional experiments were conducted on both NS-PwC and NS-Tracer-PwC, with updated results included in the revised manuscript.
>
> | Model variant                | NS-PwC  | NS-Tracer-PwC|
> | ---------------------------- | -----:  | ------------:|
> | Full DRIFT-Net (ours)        | **1.09**| **4.19**     |
> | w/o Low-Frequency Mixing     | 1.65    | 6.32         |
> | w/o Radial Gating            | 1.70    | 6.55         |
> | w/o Frequency-Weighted Loss  | 1.36    | 5.36         |
> | w/o 3×3 depth-wise conv      | 1.11    | 4.36         |
> | w/o 1×1 pointwise linear     | 1.31    | 4.21         |
>
> These ablations show that removing any main component consistently increases the relative L1 error, which empirically supports our architecture design.
>
>
> > W2: Figure 2 and Figure 3 do not seem very meaningful. If a model has lower onestep error, it will naturally have better step-wise rollout error.
>
> We appreciate the reviewer’s perspective on the relationship between one-step error and multi-step rollout performance. While one-step accuracy is certainly important, prior studies have shown that low teacher-forcing error does not necessarily translate into stable long-horizon rollouts. This is because training with ground-truth inputs but evaluating under closed-loop predictions introduces distribution shift and exposure bias, which may cause errors to accumulate over time even when the one-step loss is small [1,2]. In the context of neural PDE surrogates, recent work has therefore emphasized the importance of evaluating long-term stability and error growth control in addition to one-step accuracy [3,4]. We include Figures 2 and 3 to highlight these rollout dynamics, which are a central challenge in this line of research.
>
> Figure 2 and Figure 3 are designed to measure how errors accumulate during the rollout. On turbulent Kolmogorov flow, the one-step relative L1 errors of scOT and DRIFT-Net are 1.39 and 1.43, respectively, so scOT is slightly better at a single step. However, over a 100-step rollout (Fig. 2), DRIFT-Net shows clearly lower long-horizon errors and a smaller error-growth slope, and its rollout curve stays below scOT’s across the horizon.
>
> To make this clearer, we have augmented in the revised manuscript (Fig. 3) plots of the error growth rate over time, computed as the discrete difference of the mean relative L1 error between consecutive steps. These plots show that DRIFT-Net effectively suppresses error growth for most of the rollout.
>
> References
>
> [1] S. Bengio et al. “Scheduled Sampling for Sequence Prediction with Recurrent Neural Networks.” NeurIPS, 2015.
>
> [2] P. R. Vlachas and P. Koumoutsakos. “Learning on predictions: Fusing training and autoregressive inference for long-term spatiotemporal forecasts.” Physica D: Nonlinear Phenomena, 470, 134371, 2024.
>
> [3] P. Lippe et al. “PDE-Refiner: Achieving Accurate Long Rollouts with Neural PDE Solvers.” NeurIPS, 2023.
>
> [4] Z. Ye et al. “Recurrent Neural Operators: Stable Long-Term PDE Prediction.” arXiv:2505.20721, 2025.

---

> ### Author Response · Authors · 2025-11-24
> **Rebuttal for Reviewer Kc8E (part2)**
>
> > W3: Some other related baselines could be considered. E.g., U-Net with ConvNeXt blocks.
>
>
> We thank the reviewer for the helpful suggestion. In the revised manuscript, we have incorporated the proposed baseline: U-Net with ConvNeXt blocks (“U-Net–ConvNeXt”), into the main experimental comparison in Table 1. Detailed architectural specifications and all training hyperparameters are now provided in the Appendix F.
>
> Across all PDE tasks reported in Table 1, DRIFT-Net achieves  consistently lower final relative L1 error compared with U-Net–ConvNeXt. In addition, we have further expanded Table 1 by including additional baselines and newly added PDE datasets (highlighted in bold) to strengthen the comprehensiveness of our evaluation.
>
> | Task                | scOT | FNO   | **U-NO**  | **U-Net–ConvNeXt** | **Refiner U-Net** | DRIFT-Net |
> |---------------------|------|-------|-----------|--------------------|-------------------|-----------|
> | NS-SL               | 3.96 | 3.69  | 3.57      | 3.89               | 3.48              | **3.40**  |
> | NS-PwC              | 2.35 | 4.57  | 1.62      | 2.22               | 1.95              | **1.09**  |
> | NS-Tracer-PwC       | 5.18 | 9.46  | 10.15     | 9.63               | 4.59              | **4.19**  |
> | FNS-KF              | 4.65 | 4.43  | 11.70     | 5.94               | 4.38              | **4.32**  |
> | **Poisson–Gauss**   | 0.87 | 1.29  | 1.76      | 1.92               | –                 | **0.86**  |
> | **Allen–Cahn**      | 1.10 | 1.53  | 1.27      | 1.61               | 1.11              | **1.03**  |
> | **Wave–Gauss**      | 12.34| 16.97 | 13.94     | 16.02              | 13.33             | **12.15** |
>
> As shown in Table 1, DRIFT-Net achieves the best performance on most tasks.
>
> ### Answers to Questions
>
> > Q1: Figure 1 – is the convolution applied after the rFFT2?
>
> We thank the reviewer for the careful check. In our implementation, convolutions are applied only to spatial-domain feature maps before the rFFT2 transform; no convolutions are performed on tensors after rFFT2. As shown in Figure 1, the image branch operates purely in the spatial domain and runs in parallel with the spectral branch, after which the outputs of the two branches are combined. To avoid any ambiguity, we will refine the visual presentation of Figure 1 in the revised manuscript.
>
>
> > Q2: How does DRIFT-Net compare to scOT in terms of overall design choices?
>
> We thank the reviewer for the question. DRIFT-Net and scOT share the same high-level architecture: both adopt a U-Net–style multi-scale encoder–decoder backbone with patch merging/expansion and time conditioning, and both are trained under the POSEIDON protocol. The key distinction lies in the design of the per-layer block.
>
> In scOT, each block is SwinV2-style, where windowed self-attention serves as the primary token-mixing operation. In contrast, DRIFT-Net retains the backbone but replaces this attention block with our DRIFT block, which consists of a ConvNeXt-style image path in the spatial domain and a spectral path with rFFT2, low-frequency mixing, and radial band gating, and then sums the two paths in the spatial domain. Under this matched setup, DRIFT-Net uses slightly fewer parameters and achieves higher throughput and lower final relative L1 errors than scOT on the Navier–Stokes benchmarks.

---

> ### Author Response · Authors · 2025-11-26
> **Follow-up for Paper 18205**
>
> Dear Reviewer Kc8E,
>
> As the discussion period is entering its final week, I wanted to check in and make sure we have fully addressed your concerns. If there are any additional points or clarifications you’d like us to consider, please let us know. We truly appreciate your insights and are happy to address any remaining issues.
>
> Thank you again for your time and effort in reviewing our work.

---

### Author Response · Authors · 2025-11-24
**Summary for the Area Chair: Key Feedback and Our Revisions**

Dear Area Chair and Reviewers,

We thank the area chair and all reviewers for their thoughtful evaluations and constructive feedback.

**Reviewer Kc8E** found the presentation clear and easy to follow, highlighted that the model design reflects the motivation about improved frequency processing, and noted improved accuracy and efficiency over FNO and scOT.

**Reviewer DhGn** wrote that DRIFT-Net tackles one of the core challenges in neural PDE solvers and that the dual-branch spectral–spatial design and frequency-aware loss are sensible and well-motivated, supported by a broad experimental evaluation.

**Reviewer frHQ** emphasised that the solution to the global coupling problem intuitively makes sense and that experiments clearly show improvement over strong baselines.

**Reviewer RWgr** appreciated that  our bandwise radial gating is an effective new component which reduces accumulation errors with fewer parameters and higher throughput.

---

## Addressing the main concerns

Crucially, the remaining concerns focus on (i) positioning with respect to existing neural operators and PDE foundation models, (ii) the breadth and fairness of experiments and baselines, and (iii) the clarity and completeness of implementation details. In the revised paper, we addressed them as follows.

### Methodology and positioning

- Neural-operator taxonomy and PDE foundation models.
  We rewrote Section 2 to organise related work into DeepONet-type, integral-kernel, and transformer-based neural operators, and we now state explicitly that DRIFT-Net is a spectral–spatial integral-kernel backbone that can replace windowed-attention blocks in multi-scale operator transformers. We also added discussion of recent PDE foundation models (MPP, DPOT, ICON, CoDA-NO) and clarified that DRIFT blocks are complementary modules that could be used in such frameworks, while making explicit the current scope (uniform 2D grids, fixed POSEIDON protocol, no large-scale pre-training).

  → Revised paper: Section 2 (Related work and positioning).

### Experiments

- Expanded benchmark suite and stronger baselines.
  We increased the benchmark suite from four to seven tasks by adding Poisson–Gauss (elliptic), Allen–Cahn (reaction–diffusion), and Wave–Gauss (wave equation). We also introduced three stronger baselines—U-NO, U-Net–ConvNeXt, and a Refiner U-Net adapted from PDE-Refiner—trained under exactly the same data, optimisation, and rollout protocol as DRIFT-Net.

  → Revised paper: Section 5 (Experimental section and Table 1).

- Long-horizon behaviour, cost, and scaling.
  We designed the long-horizon Kolmogorov experiments using two ApeBench-generated variants with closed-loop rollouts up to \(T = 100\), and we now report error–time curves, error-growth slopes, and instantaneous error-growth plots. We further added width-scaling and cost tables, learning-rate and schedule sensitivity, sample-efficiency curves, cross-resolution tests, and FFT-based resolution-scaling experiments to clarify robustness and accuracy–efficiency trade-offs.

  → Revised paper: Section 5 (Long-horizon behaviour; efficiency comparison); Appendices G.1–G.5.

### Reproducibility and implementation details

- Centralised experimental protocol and configurations.
  To address concerns about missing training details and fairness of comparisons, we created a new appendix that collects, in one place, all dataset descriptions and splits, rollout horizons, variable definitions, training and optimisation settings, and exact architectural configurations for DRIFT-Net and all baselines (including scOT, FNO, U-NO, U-Net–ConvNeXt, and Refiner U-Net).

  → Revised paper: Appendix F (Experimental setup and hyperparameters), cross-referenced from Section 5.

We hope that these clarifications, new experiments, and revisions address the reviewers’ main concerns and make the contributions and limitations of DRIFT-Net clear to the AC and reviewers.

---

### Author Response · Authors · 2025-11-30
**Summary for Area Chair: Main Reviewer Concerns and Our Responses**

For each reviewer, we summarise their main concerns and how the revision addresses them.
## Reviewer Kc8E
### Main concerns
1. Ablations did not cover all DRIFT-Net branches.
2. Long-horizon rollout plots seemed redundant relative to one-step error.
3. Missing strong convolutional baseline (U-Net–ConvNeXt).
4. Asked for clarification on architectural details, including whether convolutions occur after rFFT2 and how DRIFT-Net’s design differs from scOT.
### Rebuttal
1. Performed a more complete ablation study as requested.
2. Extended the long-horizon analysis with error-growth curves and clarified where one-step error and rollout stability disagree.
3. Added a U-Net–ConvNeXt baseline.
4. Clarified that convolutions are applied only in the spatial domain before rFFT2 and explicitly contrasted DRIFT blocks with scOT’s blocks.
---
## Reviewer DhGn
### Main concerns
1. Experimental comparison was not as broad as POSEIDON.
2. No model-size scaling study.
3. Missing detailed computational-cost comparison.
4. Missing PDE-Refiner–style baseline.
5. Hyperparameter fairness and sensitivity to learning rate and schedule were unclear.
6. Whether considered evaluating POSEIDON-style sample-efficiency curves.

### Rebuttal
1. Added stronger baselines and three additional PDE tasks.
2. Added experiments training both scOT and DRIFT-Net at multiple width factors.
3. Reported parameters, peak memory, training time per epoch, and inference throughput for both models in appendix.
4. Added a Refiner U-Net baseline adapted from PDE-Refiner baseline.
5. Added new experiments on learning rates and schedules, noting that the optimisation setup comes from scOT.
6. Explained that our main metric differs from POSEIDON’s efficiency metric so such curves are less central in our setting, but still added a data-scaling study on FNS-KF.
---
## Reviewer frHQ
### Main concerns
1. Improvements over attention baselines on the long-horizon Kolmogorov datasets looked modest.
2. Theoretical justification for the model was not provided.
3. PDE scope was narrow (mainly 2D Navier–Stokes).
4. Low-frequency cutoff appeared hand-tuned.
5. Scaling behaviour with FFTs on larger grids was unclear.
6. Training settings and hyperparameters were not documented in enough detail.

### Rebuttal
1. Clarified that this benchmark primarily targets long-horizon stability and added error–time and instantaneous error-growth plots.
2. The theoretical details are provided in Appendices A–C.
3. Extended experiments to Poisson–Gauss, Allen–Cahn, and Wave–Gauss.
4. Clarified that this is listed as a limitation and future work, and discussed potential solutions to reduce reliance on a hand-tuned cutoff.
5. Added a resolution-scaling experiment.
6. Centralised all training details in appendix.
---
## Reviewer RWgr
### Main concerns
1. Asked for broader evidence of modularity and reusability beyond the current backbone and mainly 2D Navier–Stokes, and noted that other PDE foundation models were only briefly mentioned.
2. Pointed out that it does not provide sufficient evidence for a direct causal link between windowed self-attention and rollout drift.
3. Asked for a clearer accuracy–throughput comparison at matched parameter counts.
4. Noted that, without cross-resolution experiments, the paper does not yet provide evidence for discretisation invariance as in the neural-operator definition.
5. Pointed out that the baseline set, especially FNO-style variants, and the description of the FNO configuration could be expanded.
6. Asked how the design could be efficiently extended to irregular domains and complex boundary conditions.
### Rebuttal
1. Extended PDE coverage, added discussion of other PDE foundation models, and clarified that other backbones and 3D are left for future work.
2. Revised the wording, while noting that under the shared POSEIDON setup replacing windowed attention with DRIFT blocks empirically reduces error amplification and drift.
3. Compared DRIFT-Net and scOT at matched widths and reported cost metrics under fixed hardware.
4. Added cross-resolution experiments where one DRIFT-Net trained at a single resolution is evaluated at multiple resolutions.
5. Expanded the baseline set, documented the FNO configuration in more detail, and discussed related FNO variants in the revised related-work section.
6. Outlined how DRIFT-Net could be extended to irregular domains and complex boundary conditions, and clarified that this is future work.
---
## Overall
Across all reviews, the main concerns focused on:
1. Coverage and fairness of baselines and datasets
2. Completeness of ablations and theoretical support
3. Computational cost, scaling, and training setup
4. Scope and generality of our claims

The revised manuscript addresses these points with new experiments, stronger baselines, clearer theory, detailed cost and scaling studies, additional tasks, and fully documented training and implementation details.

---

### Meta-Review · Area_Chair_7wky · 2025-12-30

**Summary:**

The paper argues that windowed self-attention, as used in the scOT backbone of the Poseidon PDE foundation model and other related works, might result in long-rollout drifts for time-dependent PDEs. Therefore, the authors propose a fix in the form of DRIFT-net, a dual spectral-spatial branch neural operator, where the branches are fused. The spectral branch handles frequency domain whereas the spatial branch modulates spatial response. The authors presented extensive empirical evidence that their approach improves performance over strong baselines.

Most of the reviewers have acknowledged the strengths of the paper and the AC agrees with their judgement that the problem tackled is important and the method that they propose has been very extensively evaluated. The contribution in this regard is significant.

On the other hand, the reviewers have also pointed out some weaknesses that the authors have addressed during the rebuttal period. In particular, further ablations and additional baselines have improved the positioning of the paper.

Additionally, the AC feels that the improvements in performance over scOT are fairly marginal, particularly in the non-Navier-Stokes examples added during the rebuttal. The very long rollout gains are present but small.

Overall, the paper introduces a novel architecture and provides a consistent performance increase over strong baselines. On balance, this tends the paper towards acceptance.

**Reviewer Concerns:**

Overall, the authors have satisfactorily addressed most of the concerns of the reviewers, namely:

For Reviewer Kc8E: the main concerns were i) additional ablations for DRIFT-net ii) interpretation of long-rollouts and iii) stronger convolution baselines. The authors have addressed all these concerns satisfactorily.

For Reviewer DhGn: the main concerns were i) additional PDE examples from the Poseidon setup ii) more results on computational cost iii) PDE refiner baseline and iv) Model size scaling study. Again the authors have addressed all these concerns although it would have been nice to consider PDE-refiner in the long-rollout APEbench examples.

For Reviewer frHQ, the main concerns were i) Modest gain over scOT ii) no theoretical justification iii) narrow range of PDEs iv) FFT scaling. The authors conceded pt. i) and added additional material to address the other points of concern.

For Reviewer RWgr, the main concerns were i) Limited experimental scope ii) more discussion on FNO and other foundation models iii) the issue of resolution invariance and iv) extension to 3D and irregular geometries. The authors added new experiments and provided more discussion about future work.

**Reviewer Scores:**

Reviewer Kc8E has scored the paper as Borderline Accept and given that their concerns were addressed vis a vis baselines and ablations, it is highly likely that the reviewer would have upgraded to Accept.

Reviewer DhGn has also scored the paper as Borderline Accept and their main concerns were addressed satisfactorily by the authors with regard to more experiments and baselines. The reviewer would have likely upgraded the score to Accept.

Reviewer frHQ has scored the paper as Borderline Reject. As the main criticisms regarding additional PDEs and theoretical justification were addressed, it is likely that the reviewer would have upgraded their score.

Finally Reviewer RWgr had scored the paper as a Reject. Several of their criticisms were unfair. For instance, this paper never claimed to be a foundation model and FNO has been consistently shown to be inferior in performance to scOT and other baselines. The authors addressed the scaling and resolution invariance questions satisfactorily. It is likely that given the overall positive scores of the other reviewers, this reviewer would have updated their score.

On balance and in their independent judgement, the AC considers the contributions of the paper to be interesting although the gain in performance is modest. It falls into the Borderline Accept category.

---

### Decision · Program_Chairs · 2026-01-26

Accept (Poster)